# Brain network reconfiguration for narrative and argumentative thought

Yangwen Xu [1,2✉], Lorenzo Vignali[1,2], Olivier Collignon[1,3], Davide Crepaldi[2] & Roberto Bottini[1]

Our brain constructs reality through narrative and argumentative thought. Some hypotheses argue that these two modes of cognitive functioning are irreducible, reflecting distinct mental operations underlain by separate neural bases; Others ascribe both to a unitary neural system dedicated to long-timescale information. We addressed this question by employing inter-subject measures to investigate the stimulus-induced neural responses when participants were listening to narrative and argumentative texts during fMRI. We found that following both kinds of texts enhanced functional couplings within the frontoparietal control system. However, while a narrative specifically implicated the default mode system, an argument specifically induced synchronization between the intraparietal sulcus in the frontoparietal control system and multiple perisylvian areas in the language system. Our findings reconcile the two hypotheses by revealing commonalities and differences between the narrative and the argumentative brain networks, showing how diverse mental activities arise from the segregation and integration of the existing brain systems.

[1] Center for Mind/Brain Sciences (CIMeC), University of Trento, Trento, Italy. [2] International School for Advanced Studies (SISSA), Trieste, Italy. [3] Psychological Sciences Research Institute (IPSY) and Institute of NeuroScience (IoNS), University of Louvain, Louvain-la-Neuve, Belgium. ✉email: yangwen.xu@unitn.it

Humans are thinking animals. Flows of concepts and ideas pass through our minds from time to time. These concepts and ideas are seldom in isolation; they are often sequentially connected, composed into a mental discourse, which has been called the "train of thought"[1]. Psychologists argued for decades that these complex thoughts are essentially of two natural kinds[2,3], each gluing its elements in a different manner: Narrative thought comprises a series of events, which unfold through temporal causality and characters' intentions[4]. Argumentative thought consists of a chain of propositions, which form the interlinked logical structures, according to which a conclusion is reached through progressive inferences[5].

This thought dichotomy is deeply embodied in our language. Traditional discourse analyses recognize four common macrogenres or discourse modes, i.e., description, narration, exposition, and argument[6] (see other similar classification[7–11]). These macrogenres fall into four quadrants formed by two orthogonal dimensions (Fig. 1). The first dimension is abstractness. Descriptive and narrative texts deal with concrete and specific scenarios and events, whereas expository and argumentative texts relate to abstract and general facts and propositions. The second dimension is coherence. Descriptive and expository texts describe static states; the content of each text part can be relatively independent. Narrative and argumentative texts instead trace dynamic progress; the content of each text part usually derives from the context established before. Note that descriptive details often fuse with vivid narratives, and expository information usually serves to support a proposition as an argument. Recent theories are more likely to embrace a dichotomic view of macrogenres[12,13], and empirical studies have confirmed this division by revealing behavior differences between stories and essays in various aspects,

e.g., memorization[14,15], comprehension[15], production[16], and inference generation[17].

Nevertheless, the four-quadrant framework points out the fundamental commonalities and differences between narrative and argumentative thought. On the coherence dimension, both thought modes have high values. The content at each time point relates to the context established at previous time points. On the abstractness dimension, narrative and argumentative thought sit on opposite sides of the spectrum. Theories suggest that we rely on separate cognitive faculties to construct realities in the concrete and abstract domain[3,18]. In the concrete domain, we rely more on our own experience to build the mental representation of the state of affairs (i.e., "situation model"[19]). To trace a narrative plot, we need to simulate the situations described in the texts in order to understand the characters' implied intentions and the underlying causality linking events. This cognitive process is considered crucial to empathic understanding[20]. In the abstract domain, we rely more on logical reasoning to relate the ideas and propositions. To follow an argument, we need to identify and evaluate the logical structure embedded in the use of natural language (i.e., "informal logic"[21]), e.g., supplying the missing premises and assessing the validity and strength of the argument. This cognitive process is considered crucial to critical thinking[21].

Despite the fact that both modes of thought are pervasive in our mental life, most neuroimaging studies merely focused on the neural basis of narrative thought (see reviews[22,23]). In these studies, a narrative text was usually divided into its constituent sentences, and the order of these sentences was randomized to form a sentence-scrambled version of the text. The conditions presenting the intact texts were contrasted to conditions presenting the sentence-scrambled texts. As participants could only generate a coherent narrative discourse in the intact-text condition, this contrast isolated the cognitive component of narrative thought by subtracting the linguistic processing components concerning word meaning and syntax. A meta-analysis of 12 such neuroactivation studies indicated that narratives consistently induced greater activation than sentence-scrambled texts in the anterior temporal lobe, temporoparietal junction, precuneus, and medial prefrontal cortex[24]; a set of regions that coincides with the default mode network (DMN)[25]. Instead of investigating the overall activation level, recent studies demonstrate that the DMN activity can also capture the dynamic progress in a narrative[26–28]. As it is hard to obtain an explicit event-related response model that can describe a narrative discourse, most studies used one individual's neural response to model another's by measuring the shared neural responses across participants when they were listening to the same narrative[29]. For instance, one study used the inter-subject correlation (ISC) method and found that listening to the same narrative synchronized the blood-oxygen-level-dependent (BOLD) fluctuations between the same regions of the DMN across subjects; listening to the same sentence-scrambled text did not[26]. Another study further illustrated that such cross-subject synchronization not only exists between the same regions in the DMN (i.e., ISC) but also between different regions in the DMN (i.e., the inter-subject functional connectivity, ISFC)[27]. The later finding demonstrates that regions in the DMN underlie narrative thought by coordinating with each other as a brain network.

What is the neural basis of argumentative thought? There are two hypotheses, each corresponding to an interpretation of the DMN involvement in narrative thought[30]. The content-independent hypothesis suggests that narrative and argumentative thought share the same neural basis. As both thought modes progress coherently, iteratively accumulating information over time and holding the information online over a long timescale seems equally crucial to framing a narrative and an argument.

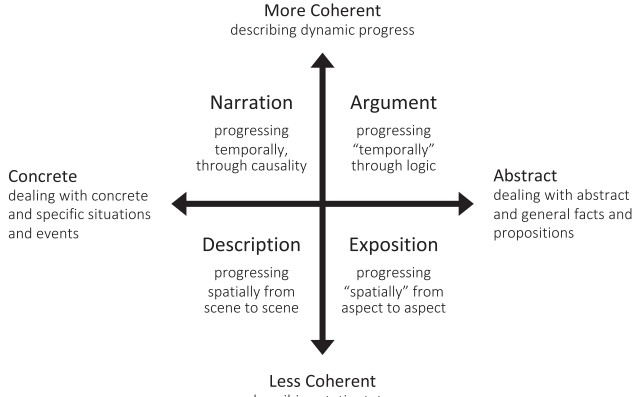

**Fig. 1 Language as a window into the thought dichotomy.** The figure illustrates the framework of four common macrogenres or discourse modes. They fall into four quadrants formed by two orthogonal dimensions, i.e., coherence and abstractness. A pure descriptive text usually progresses spatially, from one scene to another, depicting the sensorimotor and affective feelings of a scenario. A pure expository text (e.g., Wikipedia) progresses "spatially" in a metaphorical sense, from one aspect to another, expositing information about a topic. A narrative text progresses temporally, interlinking events through temporal causality and characters' intentions. An argumentative text progresses "temporally" in a metaphorical sense, framing propositions through logical relations embedded in natural language. Note that pure descriptive thought or pure expository thought rarely happens alone. Recent theories and most empirical studies are most likely to embrace a simplified dichotomic view. Nevertheless, this four-quadrant framework points out the fundamental commonalities and differences between narrative and argumentative thought.

**Table 1 Information on selected texts.**

|  |  | Narrative texts | | Argumentative texts | |
| --- | --- | --- | --- | --- | --- |
|  |  | Narrative 1 | Narrative 2 | Argument 1 | Argument 2 |
| Quoted from Book |  | Marcovaldo | The bar beneath the sea | Sapiens | The language instinct |
| Texts | N. Words | 1335 | 1158 | 1283 | 1183 |
|  | Duration (s) | 458 | 402 | 464 | 431 |
| Segments[a] | N. Segments | 58 | 50 | 54 | 52 |
|  | N. Words[b] | 23 ± 12 | 23 ± 10 | 24 ± 12 | 23 ± 11 |
|  | Duration (s)[b] | 7.9 ± 4.1 | 8.0 ± 3.4 | 8.6 ± 4.3 | 8.3 ± 4.1 |

[a]Each segment included one or more complete sentences, which ended with a period, question mark, exclamation mark, colon, or semi-colon. In the argumentative texts, each segment included one sentence. As the sentences in the narrative texts (mean ± SD: 15 ± 8 words) were on average longer than those in the argumentative texts (mean ± SD: 23 ± 11 words), in the narrative texts, each segment might include one more sentence.
[b]mean ± standard deviation.

According to the hierarchical process memory framework, all the cortical circuits accumulate information over time, but their processing timescale increases along the hierarchical topography, from milliseconds in primary sensory regions to minutes in high-order regions[31]. This framework suggests that the DMN, which is at the top of the topographical hierarchy[32,33], supports narrative thought by virtue of its wide temporal receptive window (TRW). As a wide TRW is also crucial to the progressing of argumentative thought, the DMN might potentially serve as general machinery for long-timescale information integration, supporting both narrative and argumentative thought. On the contrary, the content-dependent hypothesis suggests the two thought modes correspond to separate neural bases. Narrative thought relies on mental simulations and situation modeling. This cognitive faculty is indeed attributed to the DMN, which plays a role in experience-based simulation[25,34]. Argumentative thought, which relies on informal logic processing, should instead engage brain systems relating to language and logical reasoning.

Testing these two hypotheses requires filling the vacancy of research on argumentative thought. In this study, participants listened to two narrative texts, two argumentative texts, and their corresponding sentence-scrambled versions during the fMRI scanning. We investigated the neural correlates of both narrative and argumentative thought by contrasting the intact-text conditions to the sentence-scrambled conditions (Table 1). We also acquired the BOLD signal during the resting state as a baseline. Specifically, we employed the ISC and the ISFC as measures to respectively investigate the stimulus-induced regional activity and the interregional functional coupling during narrative and argumentative thought. The content-independent hypothesis will predict a higher ISC or ISFC in the DMN in both narrative and argumentative conditions compared to their corresponding sentence-scrambled conditions. The content-dependent hypothesis, instead, will predict a higher ISC or ISFC in the DMN only when the narrative condition and its sentence-scrambled condition are compared; alternative brain networks relating to language and logical reasoning will engage in the discourse-level comprehension of argumentative texts.

**Results**

**Behavior rating on stimuli.** Table 1 shows the information on the two selected narrative texts and the two selected argumentative texts. Each narrative text comprised one complete and independent story, and each argumentative text comprised one integrated and self-contained set of propositions in supporting a conclusion (see "Methods" for more detailed information). These texts were divided into segments consisting of complete sentences, ending with a period, question mark, exclamation mark, colon, or semi-colon. We sorted the segments according to random order and concatenated them together to generate a sentence-scrambled version for each text. The number of words, duration, the number of segments, the number of words of each segment, and the duration of each segment were matched between narrative texts and argumentative texts. These measurements were also comparable to those in the previous studies using the ISC[26] and ISFC[27] methods.

At the stimulus-selection stage, we rated narrative- and argument-relevant features of these texts on a five-point Likert scale (Supplementary Fig. 1). The questionnaire used to query these features can be found in the Supplementary Note 1. Each text was rated by 20 participants who did not participate in the MRI experiment (see "Methods" for more detailed information). The results show that the two narrative texts had significantly higher rating than the two argumentative texts on narrative-related features such as narrativeness (Welch's $t(77.81) = 20.11$; $P < 0.001$), concreteness (Welch's $t(69.93) = 3.39$; $P = 0.001$), scene construction (Welch's $t(52.52) = 9.24$; $P < 0.001$), self-projection (Welch's $t(68.92) = 5.18$; $P < 0.001$), and theory of mind (Welch's $t(77.97) = 3.99$; $P < 0.001$). The two argumentative texts had significantly higher rating than the two narrative texts on argument-related features such as argumentativeness (Welch's $t(78.00) = -10.36$, $P < 0.001$), abstractness (Welch's $t(78.00) = -11.51$, $P < 0.001$), and logical thinking (Welch's $t(77.81) = -11.03$, $P < 0.001$).

The 16 participants who took part in the fMRI experiment filled in the same rating questionnaire after scanning. The results largely validated the above rating patterns (Fig. 2). The narrative texts had significantly higher rating than the argumentative texts on the items of narrativeness (paired $t(15) = 16.37$; $P < 0.001$), scene construction (paired $t(15) = 11.28$; $P < 0.001$), self-projection (paired $t(15) = 6.92$; $P < 0.001$), and theory of mind (paired $t(15) = 4.75$; $P < 0.001$). The only exception was the rating on concreteness (paired $t(15) = 1.00$; $P = 0.331$). The argumentative texts had significantly higher rating than the narrative texts on the items of argumentativeness (paired $t(15) = -9.76$; $P < 0.001$), abstractness (paired $t(15) = -9.97$; $P < 0.001$), and logical thinking (paired $t(15) = -10.73$; $P < 0.001$). In the questionnaire, these participants also rated to which degree they understood the texts on a five-point Likert scale. The results showed that they understood the intact texts better than the sentence-scrambled texts: The comprehensibility rating on the intact narrative texts (mean ± SD: 4.69 ± 0.51) was significantly higher than the scrambled narrative texts (mean ± SD: 2.63 ± 0.67) (paired $t(15) = 15.17$, $P < 0.001$), and the comprehensibility rating on the argumentative texts (mean ± SD: 4.41 ± 0.74) was significantly higher than the scrambled argumentative texts (mean ± SD: 2.97 ± 0.99) (paired $t(15) = 8.46$, $P < 0.001$).

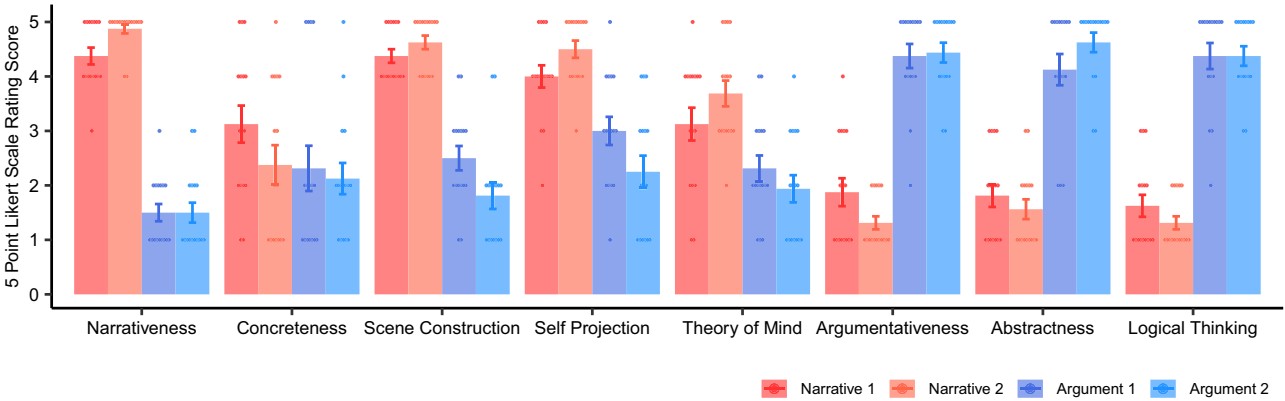

**Fig. 2 Behavior rating of the four selected texts.** The figure shows the four selected texts' rating scores on eight items from the 16 participants who participated in the fMRI experiment after the scanning. The dots denote the rating scores of individual participants, the bars denote the mean rating score across participants, and the error bars denote the standard error of the mean rating score. The narrative texts had a significantly higher rating than the argumentative texts on narrativeness, scene construction, self-projection, and theory of mind. The argumentative texts had a significantly higher rating than the narrative texts on argumentativeness, abstractness, and logical thinking. The results largely validated the rating pattern from an independent group of participants at the stimulus selection stage, as shown in Supplementary Fig. 1. In this group, the narrative texts also had a significantly higher rating than the argumentative texts on concreteness.

**Narrative, not argumentative texts, evoked time-locked neural activity in the DMN.** We first investigated the time-locked regional activity evoked by narrative and argumentative thought by comparing the ISC in the intact-text conditions when the participants could construct coherent thoughts to the ISC in the scrambled-sentence conditions when participants could only process the literal meaning of each sentence (Fig. 3). To recognize which brain systems were engaged in narrative and argumentative thought, we calculated the percentage of significant brain areas (i.e., the number of vertexes) that fell into each pre-identified brain system. The distribution of each brain system was demarcated by a brain atlas based on the clustering analysis on the interregional resting-state functional connectivity (RSFC) pattern[35] (Supplementary Fig. 2a; see "Methods" for details). We also complemented this atlas-based approach by defining the DMN using the resting-state data from the participants in the current experiment. The DMN was traced using seed-based RSFC from the posterior cingulate cortex (PCC), a core region in the DMN (Supplementary Fig. 2b; see "Methods" for details). These two approaches led to a highly similar territory of the DMN.

As a sanity check, we examined the contrast between the scrambled-sentence condition and the resting-state condition. We predicted that sentence-scrambled texts should mainly synchronize the auditory, language, and domain-general processes across participants. The results confirmed this prediction by showing that, independently of text type (narrative or argumentative), about 90% of significant vertexes fell into the four brain systems relating to auditory, language, control, and attention ($P < 0.05$, FDR corrected, area >200 mm²; Fig. 3, the first row; Supplementary Fig. 3, the first row).

We moved on to investigate the neural correlates of narrative and argumentative thought by detecting the regions that show additional or higher synchronization in the intact-text condition compared to the scrambled-sentence condition ($P < 0.05$, FDR corrected, area >200 mm²; Fig. 3, the second and third row; Supplementary Fig. 3, the second and third row). The results contrasting intact-narrative condition to the resting-state condition showed a much wider distribution of brain areas than the results contrasting scrambled-sentence condition to the resting-state condition. Note that, in the intact-narrative condition, 18% of significant regions fell into the DMN, whereas in the scrambled-narrative condition, this portion was less than 1%. Directly

contrasting the intact-narrative condition to the scrambled-narrative condition revealed about 90% of significant regions fell in four brain systems: the default mode, language, control, and attention, of which 38% were in the DMN. The significant regions in the DMN included the right angular gyrus (AG), bilateral areas comprising the precuneus, the PCC, and the ventral retrosplenial complex (RSC), the dorsal medial prefrontal cortex, and the middle portion of the left peri-hippocampal area. Intriguingly, contrasting intact-argumentative condition to the resting-state condition only showed brain areas confined within the brain areas that emerge when contrasting the scrambled-sentence condition to the resting-state condition. Directly contrasting the intact-argumentative condition to the scrambled-argumentative condition did not reveal any additional brain areas, even at a lower threshold ($P < 0.001$, uncorrected).

We also contrasted the ISC result of narrative thought to the argumentative one, i.e., (*Intact Narrative − Scrambled Narrative*) > (*Intact Argument − Scrambled Argument*) ($P < 0.05$, FDR corrected, area >200 mm²; Supplementary Fig. 4). The significant brain areas coincided with the results of narrative thought: 97% of the significant regions fell into the default mode, language, control, and attention systems, of which 38% were in the DMN. The opposite contrast did not reveal any regions that were more involved in argumentative thought than in the narrative one, even at a lower threshold ($P < 0.001$, uncorrected).

To validate the above results and to evaluate the cross-stimulus consistency, we repeated the analysis on each of the two narrative texts and the two argumentative texts. The results showed an overall consistency between the two texts of the same type despite the considerable difference in content and writing style. Texts of the same types induced more similar ISC patterns than texts of different types (Supplementary Fig. 5). For the two narrative texts, contrasting the intact-text condition to the scrambled-sentence condition revealed significant brain areas that mostly overlapped in the DMN, i.e., the precuneus and the posterior angular gyrus (Supplementary Fig. 6). For the two argumentative texts, the same contrast did not reveal any significant brain areas (Supplementary Fig. 7).

The above ISC analysis verifies the previous findings that the DMN engages in narrative thought[24,26], but fails to reveal the neural basis for the argumentative one. The results did not

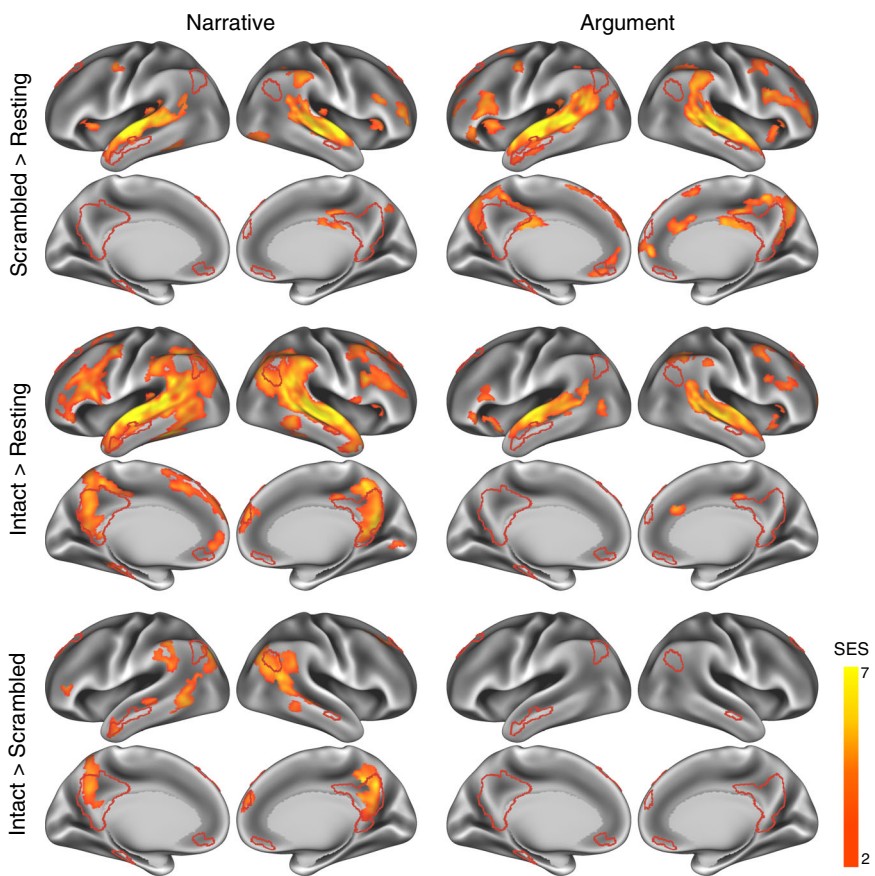

**Fig. 3 Narrative, not argumentative texts, induced time-locked neural activity in the DMN.** ISC contrast maps illustrate the significant areas of each contrast in the narrative (left) and the argumentative (right) conditions ($P < 0.05$, FDR corrected, area >200 mm$^2$). The red borderline demarcates the territory of the DMN, defined by the seed-based resting-state functional connectivity from the posterior cingulate cortex using the data from the participants in this experiment (Supplementary Fig. 2b; see "Methods" for details). The first row shows the results of the contrast between the scrambled-sentence conditions and the resting-state condition. For both narrative and argumentative conditions, mainly the auditory, language, and domain-general systems were involved. The second row shows the results of the contrast between the intact-text conditions and the resting-state condition. While the neural distribution in the intact-narrative condition extended to other brain systems like the DMN, the neural distribution in the intact-argumentative condition was confined to the areas in the scrambled-argumentative condition. The third row shows the results of the direct contrast between the intact-text condition and the scrambled-sentence condition. For the narrative condition, it only shows the areas that were also significant in the contrast between the intact-narrative condition and the resting state. Areas in the default mode, language, control, and attention systems were more engaged in the intact narratives. We did not find any significant areas in this contrast for the argumentative condition. SES, standard effect size.

support the DMN as the general machinery for long-timescale information integration, serving both modes of thought.

**Network reconfiguration for narrative and argumentative thought**. The ISC analysis investigates the stimulus-induced neural activity region by region in isolation. However, thought construction might rely on functional cooperation among regions. The ISFC, which measures the purely stimulus-induced functional coupling between discrete regions[27], can reflect brain network reconfigurations across different task states. The current analysis aimed to investigate the network reconfiguration for narrative and argumentative thought by comparing the ISFC in the intact-text conditions to the one in the scrambled-sentence conditions (Figs. 4, 5).

We implemented the ISFC analysis based on a whole-brain parcellation atlas comprising 200 brain regions[36]. The atlas also provides information about which brain system each of the 200 brain areas belongs to. Figures 4, 5 illustrate the network reconfiguration in narrative and argumentative conditions, respectively. The right panel in both figures shows the network layout of the significant ISFC differences between conditions ($P < 0.05$, FWE corrected) using the force-directed graph drawing

algorithm[37], where strongly connected nodes were clustered together, and weakly connected nodes were pushed apart. The nodes represent brain areas of each brain system, where the size of nodes denotes the node degree, i.e., the sum of edges that connect to the nodes. The edges represent the significant interregional ISFC difference between conditions, where the width of edges denotes the standard effect size (SES)[38] of the contrast (see "Methods" for the definition of the SES). The left panel in both figures summarizes the distribution of all the significant functional couplings within and between brain systems. Each cell denotes the mean SES of the contrast, i.e., the ratio of the sum of all the significant edges' SES to the number of all the possible edges in the fully connected situation.

For narrative conditions, the ISFC results were mostly in line with the ISC results. Scrambled-narrative texts, in contrast to the resting state, synchronized the neural activity mainly in the brain systems relating to auditory, language, control, and ventral attention (Fig. 4, the first row). Intact-narrative texts, in contrast to the resting state, extended the synchronization to the DMN (Fig. 4, the second row). A direct comparison between the intact- and the scrambled-narrative conditions was implemented by detecting the functional couplings that simultaneously met the

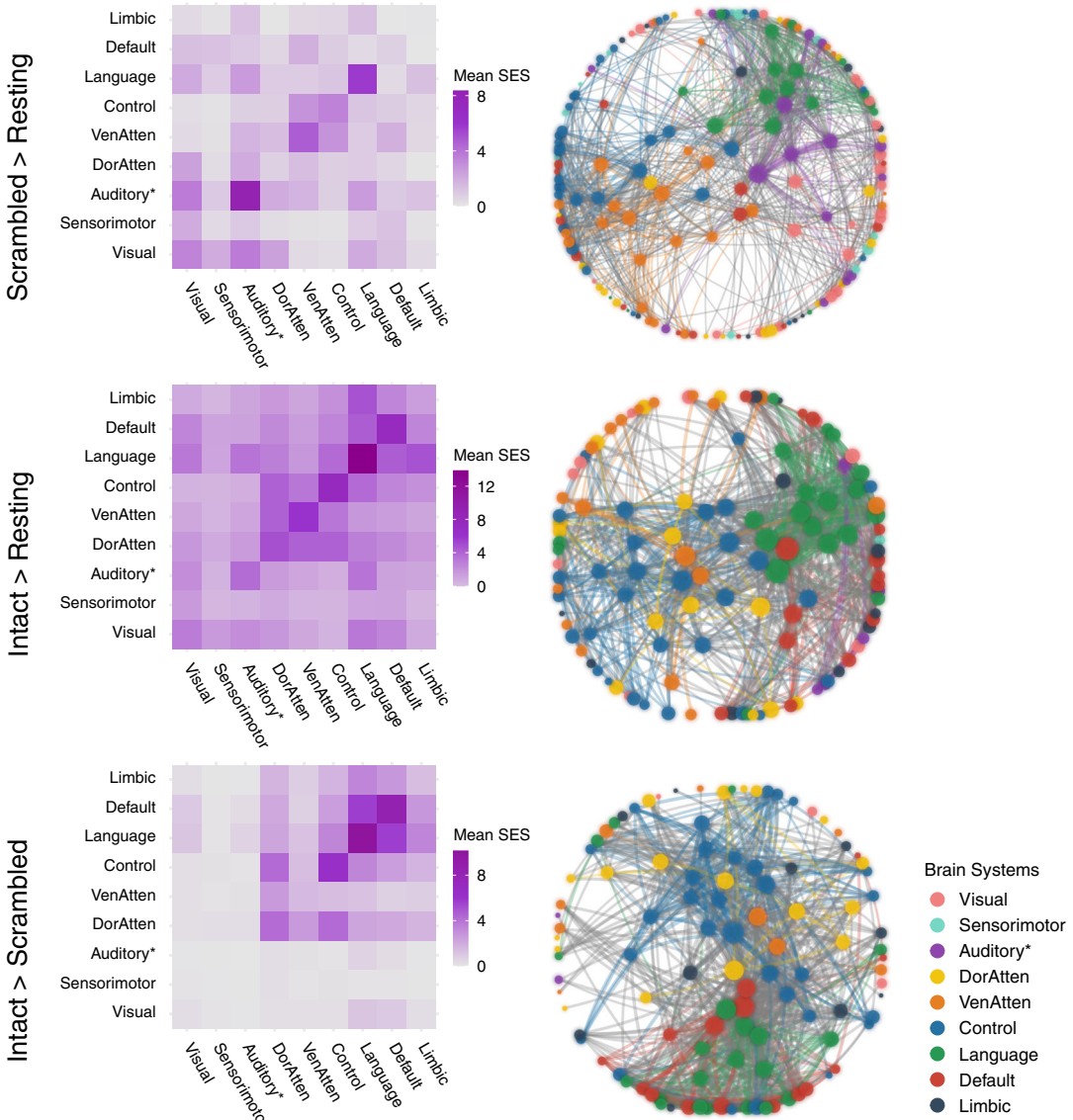

**Fig. 4 Network reconfiguration for narrative comprehension.** The figure illustrates the significant ISFC difference (*P* < 0.05, FWE corrected) between the scrambled-narrative condition and the resting-state condition (the first row), the intact-narrative condition and the resting-state condition (the second row), and the intact- and the scrambled-narrative conditions (the third row). The left column shows the distribution of all the significant functional couplings of each contrast within and between brain systems. Each cell indicates the mean standardized effect size (SES) of each contrast, i.e., the ratio between the sum of the SES and the number of the edges in the fully connected situation. The right column shows the network layout, where the nodes represent the brain areas, and the edges represent the significant interregional ISFC differences between conditions. This layout was generated using the force-directed graph drawing algorithm: strongly connected nodes are clustered together, and weakly connected nodes are pushed apart. For clarity, we just included the top 995 significant edges with the largest SES (i.e., network density equals 5%, as there were 19900 potential edges). The size of the nodes denotes the node degree of each brain area. The color of the nodes denotes to which brain system they belong. The width of the edges denotes the SES. Intra-system edges are in the color of that network; inter-system edges are in gray. "Auditory*" denotes the network including not only the auditory cortex but also the ventral somatosensory and motor brain areas corresponding to the body parts above the neck. VenAtten = ventral attention; DorAtten = dorsal attention.

criteria (1) *Intact Narrative > Scrambled Narrative* (*P* < 0.05, FWE corrected) and (2) *Intact Narrative > Resting State* (*P* < 0.05, FWE corrected). The significant edges mainly fell into the brain systems relating to the default mode, language, control, and dorsal attention. (Fig. 4, the third row). Supplementary Fig. 8a illustrates the top 20 functional couplings with the largest SES within the DMN. These critical functional couplings covered all the core regions in the DMN, i.e., the AG (Brodmann area 39), the dorsal lateral prefrontal cortex (8Ad area[39]), the medial prefrontal cortex, the PCC, the ventral RSC, and the

parahippocampal area. The result confirmed previous findings that areas in the DMN were synchronized as a network to support narrative thought[27].

For argumentative conditions, scrambled-argumentative texts, in contrast to the resting state, also synchronized the neural activity mainly in the brain systems relating to auditory, language, control, and ventral attention (Fig. 5, the first row). Intact-argumentative texts seemed not to involve additional brain systems. However, the language and the control systems became much more interconnected than in the scrambled-argumentative

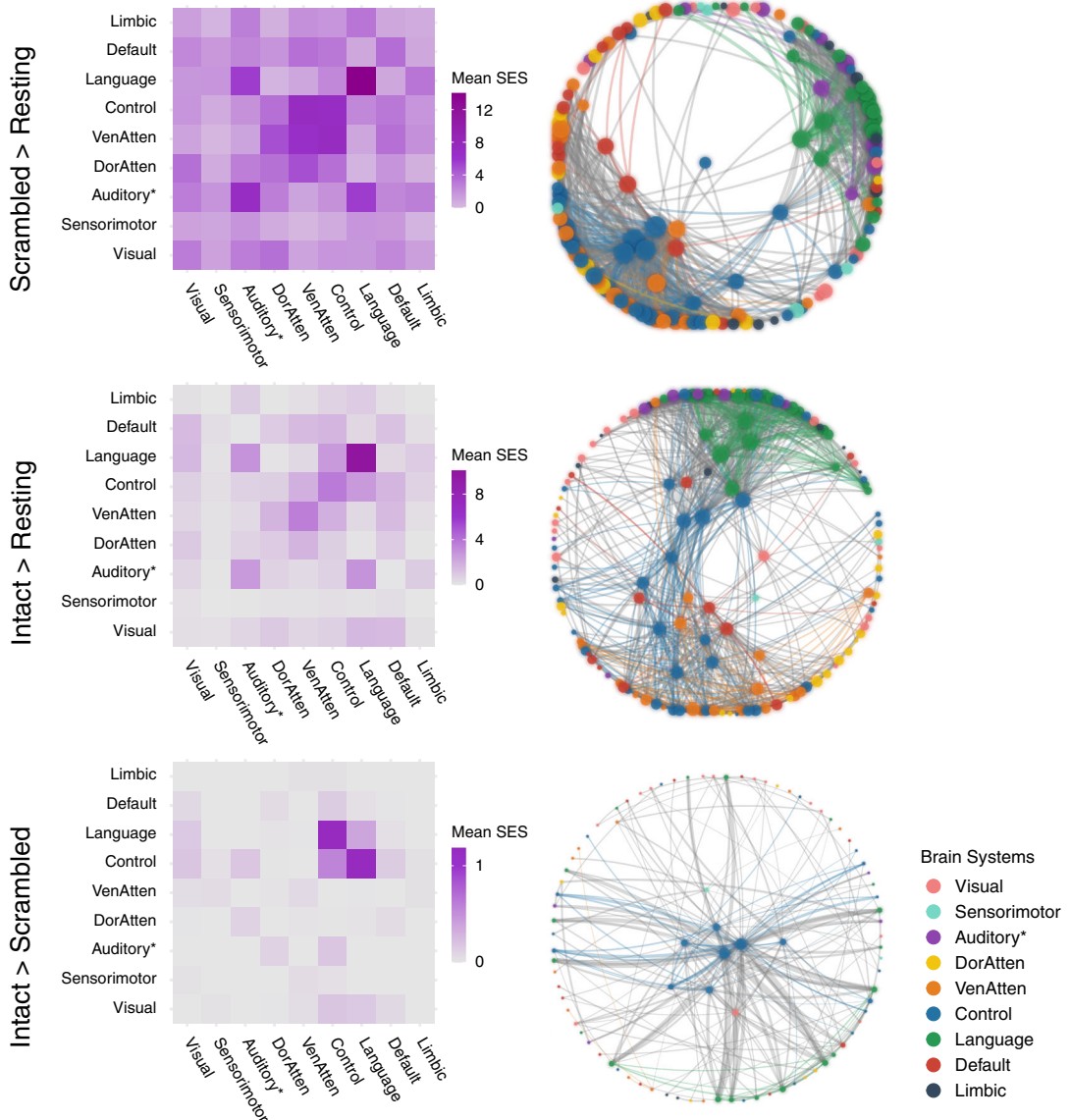

**Fig. 5 Network reconfiguration for argumentative comprehension.** The figure illustrates the significant ISFC difference ($P < 0.05$, FWE corrected) between the scrambled-argumentative condition and the resting-state condition (the first row), the intact-argumentative condition and the resting-state condition (the second row), and the intact- and the scrambled-argumentative conditions (the third row). The left column shows the distribution of all the significant functional couplings of each contrast within and between brain systems. Each cell indicates the mean standardized effect size (SES) of each contrast, i.e., the ratio between the sum of the SES and the number of the edges in the fully connected situation. The right column shows the network layout, where the nodes represent the brain areas, and the edges represent the significant interregional ISFC differences between conditions. This layout was generated using the force-directed graph drawing algorithm: strongly connected nodes are clustered together, and weakly connected nodes are pushed apart. For clarity, we only included the top 995 significant edges with the largest SES (i.e., network density equals 5%, as there were 19,900 potential edges). We included all the significant edges for the contrast between the intact- and the scrambled-argumentative conditions because there were only 217 significant edges in total. The size of the nodes denotes the node degree of each brain area. The color of the nodes denotes to which brain system they belong. The width of the edges denotes the SES. Intra-system edges are in the color of that network; inter-system edges are in gray. "Auditory*" denotes the network including not only the auditory cortex but also the ventral somatosensory and motor brain areas corresponding to the body parts above the neck. VenAtten = ventral attention; DorAtten = dorsal attention.

condition. (Fig. 5, the second row). A direct comparison between the intact- and the scrambled-argumentative conditions was conducted by detecting the functional couplings that simultaneously met the criteria (1) *Intact Argument > Scrambled Argument* ($P < 0.05$, FWE corrected) and (2) *Intact Argument > Resting State* ($P < 0.05$, FWE corrected). The significant functional couplings were mostly within the control and the language systems, especially connecting the control and the language systems. (Fig. 5, the third row). Not even a single significant functional coupling fell into the DMN. Supplementary Fig. 8b

illustrates the top 20 functional couplings with the largest SES in all the brain systems. All these critical functional couplings were between the control system and the language systems. More specifically, they were the one-to-many connections from the bilateral anterior bank of the intraparietal sulcus (IPS) in the control system to multiple perisylvian areas in the language system, including the orbital frontal cortex (Brodmann area 47), the dorsal lateral part of the temporal pole, the whole length of superior temporal gyrus/sulcus (STG/STS, Brodmann area 22), and the temporoparietal junction (TPJ).

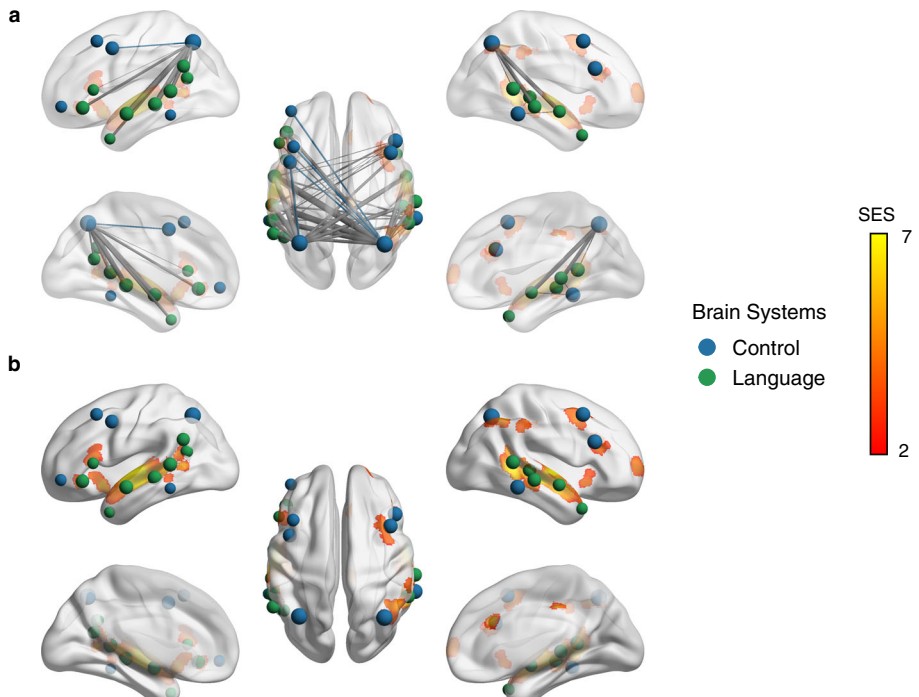

**Fig. 6 Consistency between the ISC and the ISFC results for argumentative texts comprehension.** This figure shows top 40 significant ISFC with the largest effect size in the contrast between the intact-argumentative condition and the scrambled-argumentative condition within the language and the control systems, and their overlapping with the ISC results in the contrast between the intact-argumentative condition and the resting-state condition ($P <$ 0.05, FDR corrected; cluster size >500 mm$^3$). The figure shows that the intact-argumentative condition increased the functional couplings between the regions that showed comparable ISC values to the scrambled-argumentative condition. **a** High visibility of the edges by decreasing the opacity of the brain surface. **b** High visibility of the ISC results by increasing the opacity of the brain surface. The color of the nodes denotes to which brain system they belong. The width of the edges denotes the SES. Intra-system edges are in the color of that network; inter-system edges are in gray.

Figure 6 further illustrates the correspondence between the ISC results (*Intact Argument > Resting State*; $P < 0.05$, FDR corrected) and the ISFC results (*Intact Argument > Scrambled Argument*; $P < 0.05$, FWE corrected) in the argumentative conditions. It shows that the intact-argumentative condition increased the functional couplings between the regions that showed comparable ISC values to the scrambled argumentative condition (i.e., lack of significant regions for argumentative thought in the ISC results). Compared to the scrambled-argumentative condition, the intact-argumentative condition did not involve additional brain systems but enhanced the frontoparietal functional couplings within the control system and the functional couplings between the IPS in the control system and multiple perisylvian areas in the language system. This means that argumentative thought relies on the cooperation between the control and the language systems through the connector hub—the IPS.

We also validated the above results and evaluated the inter-stimulus consistency within the same text type by repeating the analysis on each of the two narrative texts (Supplementary Fig. 9) and each of the two argumentative texts (Supplementary Fig. 10). The results indicated a substantial level of consistency between the different texts of the same type.

**Commonalities and differences between narrative and argumentative networks.** Next, we disentangled the brain network shared by both narrative and argumentative thought from the brain network specific to narrative or argumentative thought. The shared brain network for both narrative and argumentative thought was defined as the functional couplings that met the following criteria simultaneously: (1) *Intact Narrative > Resting State* ($P < 0.05$, FWE corrected); (2) *Intact Narrative > Scrambled Narrative* ($P < 0.05$,

FWE corrected); (3) *Intact Argument > Resting State* ($P < 0.05$, FWE corrected); (4) *Intact Argument > Scrambled Argument* ($P < 0.05$, FWE corrected). We found 88 functional couplings that meet these criteria (Fig. 7a). Most functional couplings were in the control system; the others were mainly within the language system or between the control and the language systems (Fig. 7b). Figure 7c illustrates the SES of the functional couplings within the control and the language systems in the contrast between each condition and the resting state. The SES in the intact condition was greater than the one in the scrambled condition for all the four texts regardless they were narrative or argumentative. Figure 7d illustrates the top 20 functional couplings with the largest averaged SES in the contrasts between the intact-narrative condition and the scrambled-narrative condition and between the intact-argumentative condition and the scrambled-argumentative condition. Most functional couplings connected areas within the control system. They connected the anterior bank of the IPS to multiple lateral prefrontal regions and the temporooccipital area at the temporal entrance (i.e., the PHT area in the Von Economo–Koskinas atlas)[40].

The brain network specific to narrative thought was defined as the functional couplings that met the following criteria simultaneously: (1) *Intact Narrative > Scrambled Narrative* ($P < 0.05$, FWE correction); (2) *Intact Narrative > Resting State* ($P < 0.05$, FWE correction); (3) nonsignificant in the contrast between *Intact Argument* and *Scrambled Argument* (uncorrected $P > 0.05$). We found 2224 edges that met these criteria (Fig. 8a). These edges are mainly related to the language, default mode, control, and dorsal attention systems (Fig. 8b). There were 87 functional couplings in the DMN. Figure 8c illustrates the SES of these functional couplings in the contrast between each condition and the resting state. The SES in the intact-narrative conditions was greater than the one in the scrambled-narrative conditions.

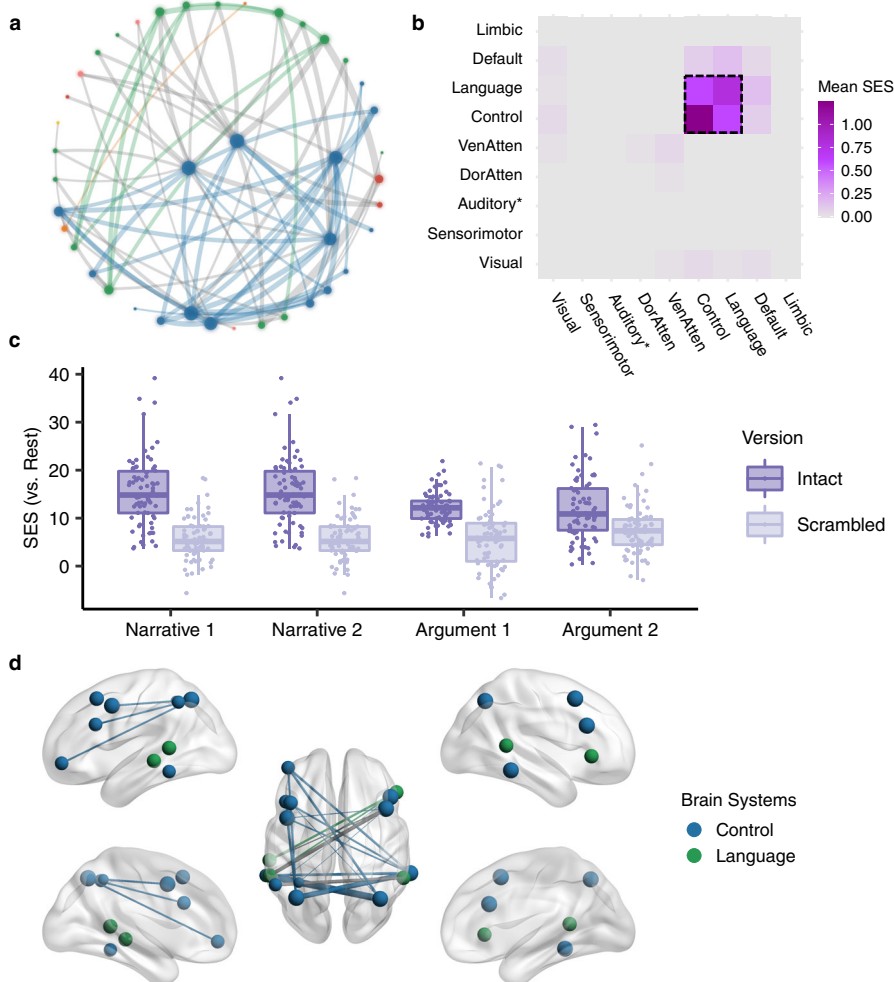

**Fig. 7 The shared network for both narrative and argumentative thought. a** The network layout of the shared brain network for narrative and argumentative thought using the force-directed graph drawing algorithm. It consisted of 88 edges. The color legend is the same as the one in Figs. 4, 5. **b** The distribution of 88 edges within and between brain systems, where each cell indicates the mean standardized effect size (SES) in the contrast between the intact and the scrambled conditions, i.e., the ratio of the sum of the SES to the number of the edges in the fully connected situation. As highlighted in the dotted box, most of the significant edges were within the control and the language systems. **c** The SES of the edges of all the conditions in contrast to the resting state within the control and the language systems. **d** The top 20 edges with the largest SES in the contrast between the intact and the scrambled conditions within the language and the control systems. In **a**, **d** the size of the nodes denotes the node degree of each brain area in the whole graph comprising 88 edges. The color of the nodes denotes to which brain system they belong. The width of the edges denotes the SES. Intra-system edges are in the color of that network; inter-system edges are in gray.

However, the SES in the intact-argumentative conditions was not greater than the one in the scrambled-argumentative conditions. Figure 8d illustrates the top 20 functional couplings in the DMN with the largest SES in the contrast between the intact- and the scrambled-narrative conditions. These edges covered all the core regions in the DMN, including the AG (Brodmann area 39), the dorsal lateral prefrontal cortex (8Ad area[39]), the medial prefrontal cortex, the PCC, the ventral RSC, and the parahippocampal area.

We also defined the brain network specific to narrative thought based on a direct comparison of narrative thought to the argumentative one. The criteria were set as (1) (*Intact Narrative − Scrambled Narrative*) > (*Intact Argument − Scrambled Argument*) ($P < 0.05$, FWE correction); (2) *Intact Narrative > Scrambled Narrative* ($P < 0.05$, FWE correction); (3) *Intact Narrative > Resting State* ($P < 0.05$, FWE correction). A total of 2348 edges met the three criteria simultaneously; 96 of them were in the DMN. These edges also covered the core regions of the DMN and were largely identical to those previously defined (Supplementary Fig. 11).

The brain network specific to argumentative thought was defined as the functional couplings that met the following criteria simultaneously: (1) *Intact Argument > Scrambled Argument* ($P < 0.05$, FWE correction); (2) *Intact Argument > Resting State* ($P < 0.05$, FWE correction); (3) nonsignificant in the contrast between *Intact Narrative* and *Scrambled Narrative* (uncorrected $P > 0.05$). We found 78 functional couplings that met these criteria (Fig. 9a). These edges mainly connected the control and the language systems (Fig. 9b). Figure 9c illustrates the SES of the functional couplings within the control and the language systems in the contrast between each condition and the resting state. The SES in the intact-argumentative conditions was greater than the one in the scrambled-argumentative conditions. However, the SES in the intact-narrative conditions was not greater than the one in the scrambled-narrative conditions. Figure 9d illustrates the top 20 edges within the control and the language systems with the largest SES in the contrast between the intact- and the scrambled-argumentative conditions. Most of the edges connected the

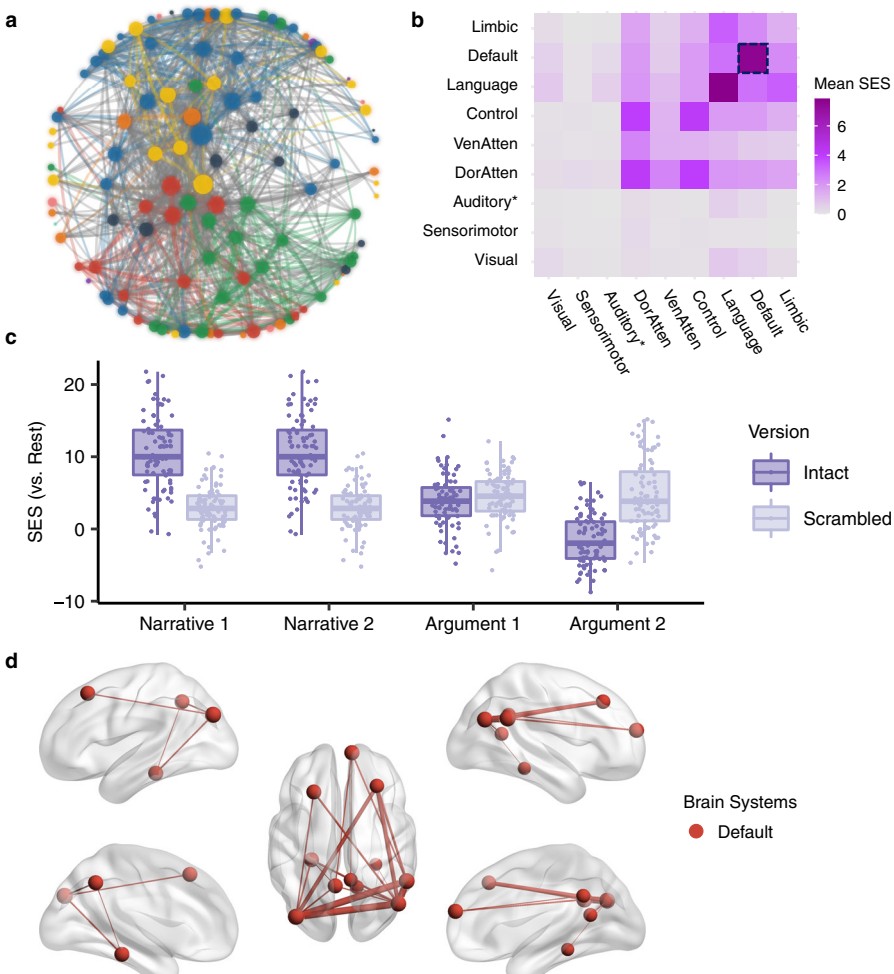

**Fig. 8 The brain network specific to narrative thought. a** The network layout of the brain network specific to narrative thought using the force-directed graph drawing algorithm. For clarity, we only included the top 995 out of 2224 significant edges with the largest standardized effect size (SES) in the contrast between the intact- and the scrambled-narrative conditions (i.e., network density equals 5%, as there were 19,900 potential edges). The color legend is the same as the one in Figs. 4, 5. **b** The distribution of all the 2224 edges within and between brain systems, where each cell indicates the mean SES in the contrast between the intact- and the scrambled-narrative conditions, i.e., the ratio of the sum of the SES to the number of the edges in the fully connected situation. There were 87 edges in the DMN, which are highlighted in the dotted box. **c** The SES of the edges of all the conditions in contrast to the resting state in the DMN. **d** The top 20 edges in the DMN with the largest SES in the contrast between the intact- and the scrambled-narrative conditions. In **a**, **d** the size of the nodes denotes the node degree of each brain area in the whole graph comprising 2224 edges. The color of the nodes denotes to which brain system they belong. The width of the edges denotes the SES. Intra-system edges are in the color of that network; inter-system edges are in gray.

control and the language systems. More specifically, they were the one-to-many connections between the bilateral anterior bank of the IPS in the control system and multiple perisylvian areas in the language system, including the orbital frontal cortex (Brodmann area 47), the dorsal lateral part of the temporal pole, the whole length of STG/STS (Brodmann area 22), and the TPJ.

We also defined the brain network specific to argumentative thought based on a direct comparison of argumentative thought to the narrative one. The criteria were set as (1) (*Intact Argument* − *Scrambled Argument*) > (*Intact Narrative* − *Scrambled Narrative*) ($P < 0.05$, FWE correction); (2) *Intact Argument* > *Scrambled Argument* ($P < 0.05$, FWE correction); (3) *Intact Argument* > *Resting State* ($P < 0.05$, FWE correction). A total of 64 edges met the three criteria simultaneously—most of them connected between the language and the control systems. These edges were largely identical to those previously defined; they were one-to-many edges connecting the IPS in the control system to multiple perisylvian areas in the language system (Supplementary Fig. 12).

## Discussion

To investigate the neural bases of narrative and argumentative thought, we compared the stimulus-evoked regional neural activity and interregional functional couplings when participants were listening to narrative and argumentative texts to those when participants were listening to sentence-scrambled texts. Our results confirmed previous findings that narrative thought engaged the brain areas in the DMN[22,24,26] and interlinked them as a network[27]. However, our results did not support that the DMN is also involved in argumentative thought. Instead, we found it was frontoparietal functional couplings within the control network that were strengthened during both narrative and argumentative thought. Argumentative thought specifically induced the functional couplings between the anterior bank of the IPS in the control system and multiple perisylvian areas in the language system, whereas narrative thought did not.

The results revealed the commonalities and differences between the neural bases underlying narrative and argumentative thought,

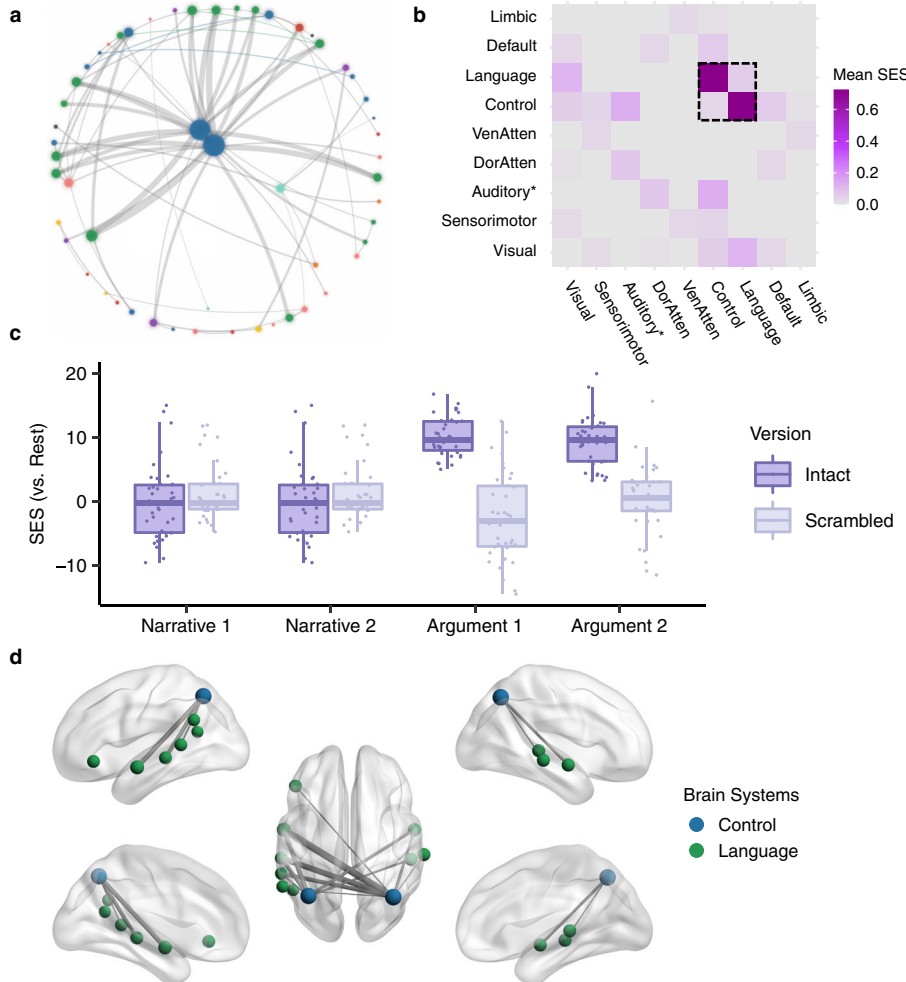

**Fig. 9 The brain network specific to argumentative thought. a** The network layout of the brain network specific to argumentative thought using the force-directed graph drawing algorithm. It consisted of 78 edges. The color legend is the same as the one in Figs. 4, 5. **b** The distribution of 78 edges within and between brain systems, where each cell indicates the mean standardized effect size (SES) in the contrast between the intact- and the scrambled-argumentative conditions, i.e., the ratio of the sum of the SES to the number of the edges in the fully connected situation. As highlighted in the dotted box, most of the significant edges were within the control and the language systems. **c** The SES of the edges of all the conditions in contrast to the resting state within the control and the language systems. **d** The top 20 edges with the largest SES in the contrast between the intact- and the scrambled-argumentative conditions in the control and the language systems. In **a**, **d** the size of the nodes denotes the node degree of each brain area in the whole graph comprising 78 edges. The color of the nodes denotes to which brain system they belong. The width of the edges denotes the SES. Intra-system edges are in the color of that network; inter-system edges are in gray.

which partially support both content-independent and content-dependent hypotheses[30]. The content-independent hypothesis predicts that narrative and argumentative thought share the same neural basis because the coherence of both thought modes relies on iteratively accumulating and updating information over a long timescale. However, instead of the DMN, as the hypothesis initially predicted in the Introduction, we found the shared neural basis for both narrative and argumentative thought in the frontoparietal control system. The frontoparietal control system, together with the attention-relevant regions in cingulo-opercular areas, is usually referred to as the "multiple demand system"[41], which is named for its broad engagement in a wide variety of demanding tasks[42]. Unlike the sustained activity in the attention-relevant brain area, the frontoparietal control system rapidly adjusts its activity profile[43] and global functional connectivity pattern[44] to adapt to the task context. Our results suggest that both thought modes may rely on the frontoparietal control system as a general working memory system to iteratively accumulating and updating information over long temporal windows.

The content-dependent hypothesis predicts that the neural bases underlying narrative and argumentative thought are irreducible to each other as these two thought modes differ in their core cognitive faculties. As mentioned in the Introduction, narrative thought relies on mental simulations to understand the implied intention of the characters and the underlying causality linking events[19]. Argumentative thought, instead, relies on informal logic processing, which includes identification and evaluation of the logic structures embedded in the natural language discourse[21]. The findings that the DMN was specific for narrative thought and the cooperation between the control and the language systems via the IPS was specific for argumentative thought may support this hypothesis. The functionality of mental simulation and situation model construction coincides with the role of the DMN in scene construction[45,46], self-projection[46,47], prospection[46,48], and theory of mind[46,49]. The role of cooperation between the frontoparietal control system and the language system in the informal logic process is also supported. Current evidence suggests the frontoparietal control system engages in

logical reasoning[50,51], whereas the language system does not[52–54], but engages in the encoding of verbal statements into the mental representations for inference operations[54]. During the argumentative text processing, the language system might be responsible for identifying the logic structures embedded in the natural language (e.g., premiss, illative, or conclusion). The encoded representations were then manipulated for logical inference by the frontoparietal control system.

Our results contradict previous findings that were interpreted as evidence to support the DMN as the content-independent network for discourse processing. For example, Ferstl and von Cramon[55] contrasted "logically" coherent sentence pairs (e.g., *Sometimes a truck drives by the house. That's when the dishes start to rattle.*) to the unconnected sentence pairs (e.g., *Sometimes a truck drives by the house. The car doesn't start.*) and still found significant areas in the DMN, e.g., the AG, the PCC/RSC, and the medial prefrontal cortex. However, this interpretation confuses "causality" and "logic." The distinction between these two concepts dates to David Hume, who points out that reasoning concerning causality is grounded in our experience, whereas reasoning concerning logically true statements is not[18]. As the study used sentences that described concrete situations, participants were most likely to use their own experience to simulate the causality linking events without resorting to logical processing.

What is the implication of the results for the functional architecture of the brain? A likely possibility is that our brain function is featured by two attributes simultaneously, i.e., the temporal receptive window (TRW) and the information types; each functional attribute is constrained by a different aspect of the brain structure. Take the frontoparietal control system and the default mode system as an example. On the one hand, according to the hierarchical process memory framework[31], the TRW of a brain system is defined by its position in the cortical hierarchy. In terms of connectivity pattern, the frontoparietal control system and the default mode system are at the middle and the top level of the cortical hierarchy, respectively[32,33]. They thus can process longer-timescale information than the sensorimotor cortices at the low level of the cortical hierarchy. On the other hand, the information type processed by a brain system is determined by its wiring patterns to the other functionally specialized brain modules. The frontoparietal control system, which has widely distributed connections to the other brain systems[56], can serve as a general machinery to integrate long-timescale information of all kinds. The default mode system, which has strong connections mainly to the medial temporal lobe, is more likely an extension to the episodic memory system[25], which serves for experience-based simulation[25,34]. Given the default mode system is at an even higher level of the cortical hierarchy than the frontoparietal control system[32,33], the default mode system could have the capacity to process longer narrative information than the frontoparietal control system, which processes domain-general information. If this is true, it might be the reason to explain why narratives tend to be more accessible and memorable than the other genres[15].

Our study also illustrates how diverse mental activities arise through network reconfiguration. There are two mechanisms at play[57]. One mechanism is through local integration. The brain was organized into functionally specialized modular structures[35]. These functional units can be selectively recruited by enhancing their within-module functional couplings according to task requirements. For example, compared to the scrambled-text condition, the intact-narrative condition selectively involved the default mode system by increasing the functional couplings among all the core regions with the DMN (Supplementary Fig. 8a, Fig. 8). The other mechanism is through global integration, which means these recruited modules are coordinated by inter-module

connections, aiming to achieve more complicated tasks. Unlike the dense intra-module connections, these inter-module connections are looser, and are usually mediated by a small number of brain areas, termed "connectors." A prominent example is the neural basis of argumentative thought. In the scrambled-argumentative condition, the language and the control systems were already involved but segregated (Fig. 5). The intact-argumentative condition did not recruit additional brain systems. Instead, it promoted the cooperation between the control and the language systems (Figs. 5–6). This cooperation was achieved strictly through the IPS as a connector (Figs. 6, 9). The global integration of the local integration strategy guarantees the efficiency and flexibility of brain function, where the functionally specialized brain modules can be combined and coordinated to adapt to diverse task contexts.

To conclude, our study revealed the commonalities and differences in brain network reconfiguration for narrative and argumentative thought. While both thought modes rely on the frontoparietal control system, narrative thought specifically implicates the DMN, and argumentative thought specifically requires the cooperation between the control and the language systems, mediated by the IPS. These results provide insights into how the brain generates diverse mental activity through global and local brain network integration.

## Methods

**Participants**. Twenty native Italian speakers who had no history of neurological or psychiatric disorders participated in the fMRI experiment. They were paid as compensation for their time. Following the experimental protocol approved by the local ethical committee at the University of Trento, all participants provided informed written consent before the start of the experiment. Data from four participants were discarded: one participant performed badly in the post-scanning questionnaire concerning the content of the narrative and the argumentative texts used in the experiment (his/her accuracy was outside 1.5 times the interquartile range below the lower quartile across participants) (Supplementary Fig. 13a). Three participants were excluded due to excessive head motion; In two cases, the mean frame displacement index[58] of functional images was outside 1.5 times the inter-quartile range above the upper quartile across participants (Supplementary Fig. 13b), and one's structure image was so blurry that it could not be segmented. The remaining 16 participants (9 females; age range: 21–31, mean age: 24) were all educated (university students or above) and right-handed (laterality quotient range: +40 to +100; mean: +90)[59]. This sample size was in line with the studies employing ISC[26] and ISFC[27] methods (11 and 18 participants, respectively).

**Stimuli**. This study employed a two (narrative vs. argumentative text) by two (intact vs. sentence-scrambled version) design. We generated two stimuli for each of these four conditions following the procedure below.

First, we searched for narrative and argumentative texts that met the following criteria: (1) written in modern Italian. (2) Easy to understand. All the texts came from best-sellers for non-expert readers. (3) Typical. The narrative text included a story with the typical elements of the story grammar[60]: settings, characters, the initial event, conflicts/goals, actions, and resolutions. The argumentative text included the interlinked premiss-illative-conclusion argumentative structure[5], with an overall conclusion at the beginning or end of the text. (4) Self-content. The narrative text should be a complete and independent story; the argumentative text should support a conclusion based on the points independent from the previous chapters. (5) Text length between 1000 and 1300 words. We posited that a comfortable speed range for an Italian audiobook is between 165 and 170 words per minute, which is slightly slower than the average speed of the *Radiotelevisione Italiana* (192.46 words per minute[61]). This criterion ensured the duration of the selected texts was relatively the same, which was about 6–8 min, comparable to the 7-min one used in the studies employing ISC[26] and ISFC[27] methods. In the end, we preselected seven such texts—three narrative and four argumentative texts.

Then, we recruited 35 native Italian speakers (who did not participate in the fMRI experiment; 11 females; age range: 23–67, mean age: 32) to rate nine features of these seven texts on a five-point Likert scale. Each participant rated four texts; hence each text was rated by 20 participants. The nine features were difficulty, narrativeness, concreteness, scene construction, self-projection, theory of mind, argumentativeness, abstractness, and logical thinking (see the questionnaire in the supplementary material). For each text, we also designed two questions on its content before the rating questions to indicate whether the participants had read and comprehended the texts (accuracy rate: 5/8 to 8/8, mean accuracy: 7/8). As all participants provided at least one correct response for each text, we did not exclude any data points. We discarded the texts with high ratings on difficulty (mean rating >3) and chose two

narrative texts and two argumentative texts as our stimuli by maximizing the difference between the ratings of these two text types: the narrative texts had higher ratings on narrativeness, concreteness, scene construction, self-projection, and theory of mind; the argumentative texts had higher ratings on argumentativeness, abstractness, and logical thinking (Supplementary Fig. 1). The two selected narrative texts came from *The wasp treatment* in the book *Marcovaldo* by Italo Calvino, who tells a story in which the protagonist asks his children to catch wasps and uses them to cure his neighbors' rheumatism (Narrative 1); *Kulala's four veils* in the book *The bar beneath the sea* by Stefano Benni, who tells a typical fairy tale (Narrative 2). The two selected argumentative texts were truncated from *Counting happiness* in the book *Sapiens: a brief history of humankind* by Yuval Noah Harari, who discusses which are the most crucial factors leading to happiness (Argument 1); *An instinct to acquire an art* in the book *The language instinct: how the mind creates language* by Steven Pinker, who argues the nature of language is an instinct faculty, not a cultural product (Argument 2).

Next, we divided the selected four texts into segments. Each segment included one or more complete sentences, which ended with a period, question mark, exclamation mark, colon, or semi-colon, i.e., we did not divide the sentences into clauses. We matched the extent of fragmentation (i.e., the number of segments and the length of segments) between these two text types (Table 1). In the argumentative text, each segment consisted of only one complete sentence. As the sentences in narrative texts (mean ± SD: 15 ± 8 words) were on average shorter than those in the argumentative texts (mean ± SD: 23 ± 11 words), in the narrative text, each segment might consist of more than one sentence.

After that, the same professional voice actor recorded all four texts with relatively the same volume, speed, voice, and tone. We cut the audio clips according to the segments that we had divided. The duration of each segment was comparable to the duration of the sentence-scrambled version (7.7 ± 3.5 s) used in the studies employing the ISC/ISFC method[26,27] and matched between the two text types (Table 1). We sorted these segments according to a random order and concatenated them together to generate a sentence-scrambled version for each text.

Finally, we added the same 10 s neutral music before both intact and scrambled versions of the stimuli following previous studies employing the ISC method[26]. The volume of the music tapered to zero before the audio texts started. As an abrupt beginning of the sound may elicit a global arousal response in the brain, a piece of opening music here helped to capture the participants' attention and to protect the start of the texts from being affected by such an arousal shift. We excluded the neural signal in this music period from the analysis (see fMRI preprocessing).

**Procedures.** Participants were told that they would be listening to the intact and the scrambled version of four texts during fMRI scanning. They were instructed to follow and comprehend the texts attentively and were informed that they would be asked to fill in a post-scanning questionnaire on the content of what they have heard. To avoid visual intrusion, we blindfolded the participants and turned off the light in the scanning room.

We presented the audio stimuli using Psychotoolbox-3 (http://psychtoolbox.org/). The sound was delivered through an in-ear headphone. Before the formal scanning, participants were instructed to check the sound in the headphone under the scanning noise. We adjusted the volume for each participant to ensure they could hear the pronunciation clearly but meanwhile did not feel too loud.

The functional scanning included nine runs, one for the eight-minute resting state, four for the sentence-scrambled version of the texts, and four for the intact version of the texts. Each task runs presented one single text. To make sure the participants were unable to replay the stimuli in the resting state, we put the resting-state run before all the task runs. To make sure the participants were unable to construct coherent thought in the sentence-scrambled runs based on the intact texts they had already heard, we put the four sentence-scrambled runs before the runs for the intact texts. The order of the four sentence-scrambled runs was randomized across participants. For the same participant, the intact-text runs followed the same order of their corresponding sentence-scrambled runs.

After the scanning, all participants completed a questionnaire on the content of the texts that they had heard during the scanning. We designed two questions for each of the four texts. In the same questionnaire, we also asked the participants to do the ratings that were used in the stimulus-selection stage. They were also asked to rate to which degree they could understand each text on a five-point Likert scale.

**MRI acquisition.** MRI data were acquired using a MAGNETOM Prisma 3T MR scanner (Siemens) with a 64-channel head–neck coil at the Centre for Mind/Brain Sciences, University of Trento. Functional images were acquired using the simultaneous multislices echoplanar imaging sequence, the scanning plane was parallel to the bicommissural plane, the phase encoding direction was from anterior to posterior, repetition time (TR) = 1000 ms, echo time (TE) = 28 ms, flip angle (FA) = 59°, field of view (FOV) = 200 mm × 200 mm, matrix size = 100 × 100, 65 axial slices, slices thickness (ST) = 2 mm, gap = 0.2 mm, voxel size = 2 × 2 × (2 + 0.2) mm, multiband factor = 5. Three-dimensional T1-weighted images were acquired using the magnetization-prepared rapid gradient-echo sequence, sagittal plane, TR = 2140 ms, TE = 2.9 ms, inversion time = 950 ms, FA = 12°, FOV = 288 mm × 288 mm, matrix size = 288 × 288, 208 continuous sagittal slices, ST = 1 mm, voxel size = 1 × 1 × 1 mm.

**MRI preprocessing.** We performed fMRI preprocessing using *fMRIPrep 1.5.0*[62], which is based on *Nipype 1.2.2*[63]. Please see Supplementary Note 2, where a boilerplate text directly generated by the *fMRIPrep* describes the preprocessing steps used in the current study. The first 10 s, which was the music period in the task runs, were labeled as the dummy scans; thus, they were excluded from the analysis. As surface-based analysis can significantly improve the spatial localization compared to the traditional volume-based analysis[64], we used the images in the fsaverge5 surface space generated by *fMRIPrep*.

We excluded the noise induced by non-neuronal sources through two steps[65]. First, we removed the motion-relevant noise using an Independent Component Analysis based strategy for Automatic Removal of Motion Artifacts (ICA-AROMA)[65]. The identified motion-relevant components and the signal components were fit into the same general linear model (GLM) to predict the BOLD signal in each vertex on the brain surface. We estimated the beta coefficients using the *fitglm* function in Matlab 2019a and subtracted the motion-relevant terms (i.e., the dot product of motion-relevant components and their estimated beta coefficients) from the BOLD signal. In this way, the motion-relevant components were removed "non-aggressively" by preserving the shared variance between the motion-relevant components and the signal components. Second, we further removed the other nuisance variables like the mean timecourses in a conservative mask of the white matter (WM) and the cerebrospinal fluid (CSF), which were extracted by *fMRIPrep*. As the low-frequency component (0–0.01 Hz) makes a significant contribution to the ISC[66], we did not implement high-pass temporal filtering but used the quadratic polynomial time trend to model the signal drift. Together, we fitted the WM timecourse, the CSF timecourse, and the quadratic polynomial time trend into the same GLM to predict the timecourse resulting from the first step. In the same way, we estimated the beta coefficients and subtracted the WM, the CSF, and the quadratic polynomial terms from the signal.

We implemented the surface smoothing on the resulting images with a full width at half maximum of 8 mm using the mri_surf2surf command in FreeSurfer (http://surfer.nmr.mgh.harvard.edu/). The timecourse in each vertex was then z-normalized across time points to enter the analyses.

**Brain network identification.** We identified the brain systems based on a pre-labeled atlas[35]. The brain systems in this atlas were identified by applying the clustering analysis on the pattern of 1000 young healthy participant's resting-state functional connectivity (RSFC). The atlas has two versions: one coarse version with seven networks and one fine-resolution version with 17 networks. We chose the fine-resolution version as the start for two reasons. First, the fine-resolution version separates the dorsal somatosensory and motor cortex corresponding to the body parts mainly below the neck from the ventral networks consisting of the auditory cortex and the somatosensory and motor cortex corresponding to the body parts mainly up the neck. This division helps us to differentiate the auditory cortex from most of the somatosensory and motor areas. Second, the fine-resolution version also separates the language network[67] and the DMN[25]. Previous studies suggest these two networks are dissociated in respective of both activation profile and functional connectivity pattern[34,68,69]. We merged Network 14 and Network 17 as the language network, which mainly includes the perisylvian cortex and the 55b area[67,70]. We merged Network 15 and Network 16 as the DMN, as these two networks largely correspond to the two identified sub-networks of the DMN[71]. We preserved the labels used in the coarse version of the atlas for the other brain networks. These networks were visual; ventral attention[72,73], which may implicate multiple networks variably referred to as the salience[74] and the cingulo-opercular[75]; dorsal attention[72,73]; frontoparietal control[75,76]; and limbic. In the end, we obtained an atlas, including nine brain systems (Supplementary Fig. 2a).

We also defined the DMN with the data specifically from the participants in this experiment. We used the PCC, a core region in the DMN, as the seed and traced the DMN by calculating the RSFC between the PCC and the rest of the brain. The PCC ROI was taken from the brain atlas as a parcel with a homogenous RSFC pattern and belonging to the DMN ("pCunPCC_1" region in the brain atlas comprising 17 networks and 400 parcels)[36]. We fisher-z transformed each participant's correlation coefficient image and implement a one-sample *t*-test across participants with PALM (Permutation Analysis of Linear Models, 10,000 permutations with sign-flipping)[77]. Significant regions were defined as the DMN (family-wise error rate < 0.001, area > 200 mm²). These regions closely resembled the DMN distribution in the pre-labeled brain atlas mentioned above[35] and covered all the core regions of the DMN[25] (Supplementary Fig. 2b).

**ISC analysis.** The ISC was defined as Pearson's correlation between the timecourse in the same area of different participants. We calculated the ISC for each vertex and each run using a leave-one-participant-out approach. For each participant, we first averaged the timecourses of all the other participants and then correlated this mean timecourse with this participant's timecourse. The resulting Pearson's correlation coefficients (one per participant) were Fisher-z transformed using the inverse hyperbolic tangent function before they were averaged as one ISC index. In this way, we obtained one ISC surface map for each of the nine runs.

We contrasted the ISC surface maps between different conditions to obtain a veritable ISC contrast value for each vertex for each contrast. The major contrasts were: (1) Scrambled Narrative Contrast: (Scrambled Narrative 1 + Scrambled Narrative 2) − 2 × Rest; (2) Intact Narrative Contrast: (Intact Narrative 1 + Intact

Narrative 2) − 2 × Rest; (3) Narrative Contrast: (Intact Narrative 1 − Scrambled Narrative 1) + (Intact Narrative 2 − Scrambled Narrative 2); (4) Scrambled Argumentative Contrast: (Scrambled Argument 1 + Scrambled Argument 2) − 2 × Rest; (5) Intact Argumentative Contrast: (Intact Argument 1 + Intact Argument 2) − 2 × Rest; (6) Argumentative Contrast: (Intact Argument 1 − Scrambled Argument 1) + (Intact Argument 2 − Scrambled Argument 2); (7) Narrative Specific Contrast: [(Intact Narrative 1 − Scrambled Narrative 1) + (Intact Narrative 2 − Scrambled Narrative 2)] − [(Intact Argument 1 − Scrambled Argument 1) + (Intact Argument 2 − Scrambled Argument 2)]; (8) Argumentative Specific Contrast: [(Intact Argument 1 − Scrambled Argument 1) + (Intact Argument 2 − Scrambled Argument 2)] − [(Intact Narrative 1 − Scrambled Narrative 1) + (Intact Narrative 2 − Scrambled Narrative 2)]. We also implemented similar contrasts using individual narrative texts and individual argumentative texts to validate our results and to evaluate the inter-stimulus consistency. The following ISFC analysis used the same contrasts here.

The statistical likelihood of each contrast was assessed using the subject-wise bootstrapping method, where the exchangeability and independence assumptions are satisfied[78]. To accommodate the within-subject data structure, we resampled the participants and calculated each sample's contrast value rather than resample the participants independently for each condition and then calculated the contrast value. In each bootstrapping iteration, the same number of participants were randomly resampled with replacement. The ISC was calculated between the timecourse of one participant and the mean timecourse of the other participants. Here, "the other participants" were those excluding him/herself and the repeated ones of him/herself due to resampling with replacement, which guaranteed that the ISC was always between participants without any overlap. The obtained Pearson's correlation coefficients (one per participant) were Fisher-Z transformed and averaged. We then contrasted these maps between conditions in the same way as before. This procedure was repeated 5000 times to form a sampling distribution for each contrast. The null distribution of each contrast was generated by subtracting the veritable contrast value from the sampling distribution, and the veritable contrast value was then ranked against the null distribution[79]. As the null distribution of each contrast of each vertex was symmetrical (the skewness is within ±1), to provide a quantitative measure of the magnitude across contrasts and vertexes, we calculated the standardized effect size (SES) as $(x − \mu)/\sigma$, where $x$ is the veritable contrast value, $\mu$ is the mean of the null distribution, and $\sigma$ is the standard deviation of the null distribution[38]. To obtain a high-resolution P-value given the limited number of resamples, we estimated the right-tail p-value of each contrast by approximating a generalized Pareto distribution to the tail of the null distribution[80]. We corrected for multiple comparisons across the entire brain surface using the false-discovery rate (FDR) correction algorithm without the need for the assumption of independence across vertices[81] ($P < 0.05$).

**ISFC analysis**. The ISFC was defined as the Pearson's correlation between the timecourse in two discrete brain areas from different participants. We defined the brain areas based on the cortical parcellation derived by integrating the local gradient approach, which detects the abrupt transitions in RSFC patterns, and the global similarity approach, which clusters similar global ISFC patterns despite the spatial proximity[36]. Thus, the obtained parcels are locally homogenous and globally match the brain networks shown above[35]. We chose the brain atlas matched to the 17 brain networks and then relabeled them as nine networks of interest. Considering the trade-off between the spatial resolution and the computational load, we chose the cortical parcellation consisting of 200 parcels. The averaged time-course across all vertexes in each parcel was used as the timecourse of that parcel.

We calculated the pairwise, inter-regional ISFC among the 200 parcels for each run using a leave-one-participant-out approach following the previous study[27]. A 200 by 200 ISFC matrix $C$ was obtained for each of the nine runs, where each element in the matrix (e.g., $C_{ij}$) represented the ISFC strength between each pair of regions (e.g., the ith and the jth brain areas). To calculate the value of $C_{ij}$, we first averaged the timecourses of all the other participants in the jth area and then correlated this mean timecourse with this participant's timecourse in the ith area. The resulting Pearson's correlation coefficients (one per participant) were then Fisher-z transformed, averaged, and assigned to $C_{ij}$. Accordingly, $C_{ji}$ referred to the correlation between the timecourse in one participant's jth area and the mean timecourse across the other participants' ith areas. As this leave-one-participant-out approach of calculation could not guarantee $C_{ji}$ equals $C_{ij}$, the ISFC between the ith area and the jth area was defined as the mean of $C_{ij}$ and $C_{ji}$. To this end, we symmetrized the ISFC matrix as $(C + C^T)/2$, where $C^T$ is the transpose of the matrix $C$. We contrasted the ISFC matrix between different conditions using the same contrasts in the ISC analysis to obtain a veritable contrast value for each pair of brain areas for each contrast.

The statistical likelihood of each contrast was assessed using a similar subject-wise bootstrapping method shown in the ISC analysis. In each iteration of the bootstrapping, the same number of participants were randomly resampled with replacement. A 200 by 200 ISFC matrix $C$ was calculated using the data from this sample, where each element in the matrix (e.g., $C_{ij}$) represented the ISFC strength between each pair of brain areas (e.g., the ith and the jth brain areas). $C_{ij}$ was calculated as Pearson's correlation coefficient between the timecourse in the ith brain area of one participant and the mean timecourse in the jth brain area of the other participants. Here, "the other participants" were those excluding him/herself and the repeated ones of him/herself due to resampling with replacement, which

guaranteed that the ISFC was always between participants without any overlap. The obtained Pearson's correlation coefficients (one per participant) were Fisher-Z transformed, averaged, and assigned to $C_{ij}$. We symmetrized the ISFC Matrix $C$ in the same way as before. We contrasted these final ISFC matrixes between conditions, ending this iteration. This procedure was repeated 5000 times to form a sampling distribution of the ISFC contrast value for each pair of brain areas for each contrast. The null distribution of each contrast was generated by subtracting the veritable contrast value from the sampling distribution[79]. As the null distribution of each contrast of each pair of brain regions was symmetrical (the skewness is within ±1), to provide a quantitative measure of the magnitude across contrasts and pairs of brain regions, we calculated the SES as $(x − \mu)/\sigma$, where $x$ is the veritable contrast value, $\mu$ is the mean of the null distribution, and $\sigma$ is the standard deviation of the null distribution[38]. We controlled the family-wise error (FWE) rate by defining the threshold at the 5% percentile of the null distribution of the maximum across all pairs of brain areas and thresholded the SES matrix by assigning the nonsignificant brain pairs to zero.

The resulting thresholded SES adjacency matrix in each contrast was modeled as a weighted graph comprising nodes and edges[82]; the nodes represented brain areas, and the edges represented the SES of that contrast for the ISFC between each pair of the brain areas. We used the node degree to measure the importance of one brain area in each contrast. The degree of the ith node was calculated as $\sum_{j=1}^{200} C_{ij}$, where $C$ was the thresholded SES adjacency matrix.

**Statistics and reproducibility**. We implemented the two-sample t-test and the paired t-test to compare the behavior rating between narrative and argumentative texts. We employed the subject-wise bootstrapping method to make statistical inferences on the ISC and ISFC analyses to make sure the exchangeability and independence assumption was satisfied[78]. We largely replicated our behavior rating results on an independent group of participants at the stimuli-selection stage and on the group participants who took part in the fMRI experiment. We replicated our ISC and ISFC results by repeating the analyses on each of the two narrative texts and each of the two argumentative texts.

**Visualization**. The ISC results were illustrated using the Connectome Workbench 1.3.2 (https://www.humanconnectome.org/software/connectome-workbench). For the visualization purpose, we mapped the significant clusters from the fsaverage5 surface to the fsLR surface using the ADAP_BARY_AREA method. We excluded the clusters that are smaller than 200 mm². The significant clusters were illustrated on an inflate surface against the group-averaged sulcus image of 1096 young adults from the dataset under the Human Connectome Project (https://balsa.wustl.edu/reference/pkXDZ).

For the ISFC results, the network layout was generated using the force-directed graph drawing algorithm[37] with NodeXL (https://www.smrfoundation.org/nodexl/)[83]. The strength of the repulsive force between nodes was set as 30. This parameter provided a better view of the internal structure among the core regions in the center, e.g., whether the control and the language systems were connected during the argumentative conditions. The byproduct was that the peripheral and less important nodes would have not enough room to be fully extended and be extruded into a circle. The brain networks were visualized with the BrainNet Viewer (https://www.nitrc.org/projects/bnv/)[84]. To localize each node, we used the centroid of the Montreal Neurological Institute (MNI) coordinates of each brain parcel in the volume version of the same brain parcellation atlas. The joint visualization of the ISC and the ISFC results (Fig. 6) was illustrated using the BrainNet Viewer by transforming the ISC results from the fsaverage5 surface space to the MNI volume space with the ribbon constrained method in the Connectome Workbench.

**Reporting summary**. Further information on research design is available in the Nature Research Reporting Summary linked to this article.

## Data availability

The datasets generated during the current study are available from the corresponding author on reasonable request. Source data underlying the figures are available in the figshare repository (https://doi.org/10.6084/m9.figshare.14433008).

## Code availability

Code for ISC and ISFC analyses is available (https://github.com/BottiniLab/NarrativeArgumentative).

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

## Acknowledgements
We thank Alessia Zampieri for the help in stimuli selection, Edoardo Camponeschi for recording the stimuli, and Anna D'Urso for the help in data collection. This research is supported by the Caritro Foundation and the Research Projects of National Interest (PRIN) grant from the Italian Ministry of Education, University and Research (MIUR) to D.C. and O.C. (project number: 2015PCNJ5F_001).

## Author contributions
X.Y. and R.B. conceived the experiment. X.Y. and L.V. analyzed the data. X.Y. wrote the paper with input from L.V., O.C., D.C., and R.B. All authors discussed the results and contributed to the paper.

## Competing interests
The authors declare no competing interests.
