## [Peer Review File · Communications Biology]

Reviewers' comments:

Reviewer #1 (Remarks to the Author):

Xu et al. used fMRI to examine commonalities and differences in functional activation and functional connectivity patterns during two different forms of thought: argumentative and narrative. The authors took advantage of inter-subject synchronization-based analyses to quantify these patterns. While both modes of thought engaged auditory regions along the superior temporal sulcus and inferior frontal gyrus, narrative thought additionally activated and synchronized default mode network regions. In contrast argumentative thought demonstrated increased synchronization of auditory and control networks. These findings point to somewhat overlapping yet distinct brain activation patterns for these modes of thought. Overall, I really like this study topic and the methods that were used. Understanding the neural correlates of different modes of thought is a key topic to address and has not yet been fully investigated. In this regard the present study will make a nice contribution to the literature as it highlights some potentially interesting differences involving the default mode network. I do have some comments that I hope will strengthen the manuscript.

I think it would be useful to have a little more detail in the definition of the two modes of thought in the first paragraph of the Intro and perhaps include a short example of each. I found myself not completely clear on the core elements that distinguish them. It might also be useful to give some context that will anticipate the questions that were asked of subjects related to abstractness, degree of scene construction, theory of mind etc. To what extent are these defining features of each mode of thought versus details that differed in the specific examples used, but may not necessarily distinguish these modes of thought in general. The authors rightly point out that the literature has mainly focused on narrative thought, so the rest of the paper really depends on having readers understand what argumentative thought is and how it differs from narrative thought.

I thought that the framing of the study was somewhat problematic. The general idea is that the study would use brain data to adjudicate between two hypotheses—that narrative and argumentative thought are fundamentally different or fundamentally the same. However, in order to test this, the authors used stimuli that were chosen specifically because they have different properties, at least from a psychological perspective. So, there is really no information that is gained from the brain in terms of quantifying how different these modes of thought are. Moreover, this framing relies on reverse inference which is a notoriously challenging approach to do correctly (and would require much more justification than is currently offered). I think that the study would be just as strong and not suffer from these issues if it were simply presented for what it is—an investigation of the neural correlates of different modes of thought. This becomes a forward inference approach which has fewer thorny issues to contend with. I don't think there is any problem in mentioning that there are different conceptions of how similar/distinct these modes of thought are. But I do see a problem in framing the study as a test of these hypotheses using brain data.

The results suggest that the DMN is really only involved in narrative thought and not argumentative thought. This is potentially a very interesting finding. However, one possibility is that the DMN is recruited during argumentative thought, but in a way that is more idiosyncratic across subjects. This possibility could be addressed by presenting results from a standard group level analysis (rather than ISC) using an intact argumentative > scrambled argumentative contrast. If this reveals DMN activation it would suggest that this mode of thought does rely on the DMN but that the precise timing of activation is not consistent across participants (hence no inter-subject synchronization). I think this would provide helpful information that could strengthen (or weaken) the claim that the DMN is specific to narrative thought.

To compare functional coupling between the two modes of thought, the authors rely on a set of criteria which include a significant contrast (e.g., intact narrative > scrambled narrative) and a non-significant contrast (e.g., intact argumentative > scrambled argumentative). However, this is generally not sufficient to demonstrate a condition difference. A direct contrast (e.g., intact narrative > intact argumentative) is necessary. Or it could be in the form of [intact narrative >

scrambled narrative] > [intact argumentative > scrambled argumentative]. Without a direct contrast it is not possible to make conclusions about the specificity of various coupling patterns.

I found Figure 3 a bit hard to interpret. The purple color bar was not very intuitive and personally, I find circular embedded graph layouts less intuitive than spring embedded graph layouts. However, I realize that these are just my subjective opinions and I completely leave it to the authors to consider alternative formats or to retain their initial choices.

Line 48: "Another study further illustrates that such higher synchronization in the DMN not only exist between..." should read "Another study further illustrated that such higher synchronization in the DMN not only exists between..."

On line 173, SES is not defined. I also found myself desiring more commonly used statistics in the text and figures such as t and r values. Is there some reason why these could not be included as well?

On line 239, I'm not familiar with the term "temporal entrance".

Line 310, I believe "medial" is meant to be "middle"

Line 327, "unite" should be "unit"

Reviewer #2 (Remarks to the Author):

In this manuscript, Xu and colleagues use the method of intersubject correlations (ISC) to examine the fMRI correlates of narrative and argumentative thoughts in adults. They found commonalities and differences in activity and connectivity between both types of discourses.

I think that the study addresses a very interesting question, using a solid methodological approach that may shed light on the brain mechanisms underlying discourse processing in different contexts. However, the manuscript also has several weaknesses and would benefit from a revision to strengthen its conclusions and clarify its contribution to the current literature. My comments are listed below.

Major issues:

1) The literature review is sparse. There is quite an extensive literature on the brain mechanisms underlying discourse processing and their relation with the default mode / Theory of Mind network (e.g., Ferstl, Neumann, Bogler, & von Cramon, 2008; Ferstl & von Cramon, 2002; Jacoby & Fedorenko, 2018; Mar, 2011; Paunov, Blank, & Fedorenko, 2019; Prado, Léone, Epinat-Duclos, Trouche, & Mercier, 2020). Although some of these studies are briefly mentioned, many are missing and there is very little discussion of this literature in either the introduction of the discussion. This is quite surprising as several of these studies have examined questions that are related to the present investigation. For example, back in 2002, Ferstl and von Cramon already looked at the neural correlates of logical inferences during discourse comprehension. I believe that the paper would benefit from a more thorough discussion of this literature, which would also clarify the contribution of the present paper with respect to these previous studies.

Similarly, the authors make assumptions about what could be expected in terms of neural bases of argumentative thought based on a very limited literature. For example, one reference to "informal logic" from a specific chapter led them to infer that, according to the content-dependent hypothesis, argumentative discourse "might warrant cooperation between the language and the reasoning brain systems". There is an extensive field in the psychology of reasoning whose goal is to determine whether logical reasoning related to natural language (e.g., see Monti & Osherson, 2012; Prado, 2018). There is also an even more relevant literature on the neuroimaging of reasoning investigating the very question of whether logical thought is supported by the

mechanism of language (e.g., (Holyoak & Monti, 2020; Monti, Parsons, & Osherson, 2009; Reverberi et al., 2007). I think the paper would benefit from a discussion of this literature, again so that the contribution of the present manuscript can be clearly fleshed out.

2) There is no direct comparison of narrative versus argumentative texts. The authors attempt to show that there are differences in the neural bases of narrative and argumentative texts by combining maps showing significant effects for one condition and insignificant effects in the other. Yet, this does not show that the effect in one condition is greater than the effect in the other. This would need to be statistically demonstrated, for example by directly comparing the contrast Intact Argument > Scrambled Argument to the contrast Intact Narrative > Scrambled Narrative (and vice versa). See Nieuwenhuis, Forstmann, & Wagenmakers (2011) for a discussion of this classic issue in neuroimaging studies.

3) The resting state data could be used to identify the DMN in each subject. At the heart of the analysis strategy is the idea to compare the brain networks subserving narrative and argumentative thought to the DMN. However, although the authors do acquire some resting state data, they only make use of these data as a control condition against which maps associated with texts are compared. Instead of comparing activity/connectivity associated with texts to general regions of the DMN defined by a brain atlas, I think it would strengthen the authors' conclusions to (a) visualize the DMN for this specific group of participants and (b) use these specific regions to compare the DMN to the network associated with the contrasts of intact versus scrambled texts. This would give more specificity to the inferences and increase confidence in the conclusions regarding overlap with the DMN.

4) The discussion relies too heavily on reverse inferences. I think the authors should keep in mind that they do not directly localize the brain regions supporting language processing, attentional processing, visual processing or even the DMN. These are simply taken from an atlas. Therefore, they should be careful not to overinterpret involvement of the different brain regions as a function of these processes. For example, on the basis on one single study, they claim that the IPS is the "neural basis of fluid intelligence" (line 302) and therefore argue that "the coordination and cooperation between the frontoparietal control system and the language system, which is mediated by the IPS, might be critical to identify and evaluate the informal logic in the natural language discourse." This is extremely speculative as one could find a long list of functions subserved by the IPS besides fluid intelligence (e.g., attention, working memory, quantitative processing etc.). The authors should stay away from these sorts of inferences (there are others in the discussion, for example regarding the role of regions in the DMN, the frontoparietal system etc.) and focus on what their study show in terms of commonalities and quantitative differences between narrative and argumentative texts. For this, a direct comparison of conditions is necessary (see point #2 above).

5) There are inconsistencies in the hypotheses. The authors argue in the introduction that two hypotheses can be made regarding argumentative and narrative thought. On the one hand, the content-dependent hypothesis posits that the DMN should only support narrative texts and not argumentative texts. On the other hand, the content-independent hypothesis posits that both texts should be supported by the DMN, which "might potentially serve as general machinery for long-timescale information integration, supporting both narrative and argumentative thought." (line 73). If I understand correctly this framework as it is laid out in the introduction, the results clearly give support to the content-dependent hypothesis as the DMN is only found in narrative texts. However, the authors appear to slightly change the hypothesis in the discussion, arguing that because there is overlap in other regions than the DMN (notably the frontoparietal network), the content-independent hypothesis is still supported (e.g., "The results revealed the commonalities and differences between neural bases underlying narrative and argumentative thought, which seems to support both content-independent and content-dependent hypotheses"). This seems inconsistent with what is stated in the introduction to me. If the theory makes the prediction that the DMN is the center of this content-independent processing, then I do not see how it is supported. The authors should clarify the hypotheses, make sure they are consistent throughout the paper and interpret their results with respect to the hypothesized framework.

6) The sample size is rather small. After excluding several participants, only 16 participants are analyzed. This seems quite small and the study might be underpowered if one considers typical effect sizes in neuroimaging studies (Poldrack et al., 2017). The only rationale given is that “the sample size was in line with the studies employing ISC and ISFC methods (11 and 18 participants, respectively).” This is not very convincing, as it is well known that most prior neuroimaging studies are largely underpowered (Button et al., 2013) and it is likely that both of these studies are also underpowered. The authors should provide a clear justification for their final sample size, ideally based on a power analysis.

Minor issues:

7) If I understand correctly, the authors use ratings from another group of participants to classify texts as narrative versus argumentative. It is then not surprising that, in this specific group, the ratings differ significantly with respect to the measures that were used to classify the texts. The results reported from line 100 to 107 are thus a bit trivial and this should be more clearly indicated. Additionally, what is less trivial is whether the participants in the fMRI experiment also differed with respect to the rating. These results are presented in Figure S1, I would move them to the main text (and also move Figure S1 to the manuscript).

8) line 506: “We calculated the ISC for each vertex each run” should be “We calculated the ISC for each vertex and each run”

Button, K. S., Ioannidis, J. P. A., Mokrysz, C., Nosek, B. A., Flint, J., Robinson, E. S. J., & Munafò, M. R. (2013). Power failure: why small sample size undermines the reliability of neuroscience. *Nature Reviews. Neuroscience*, 14(5), 365–376.

Ferstl, E. C., Neumann, J., Bogler, C., & von Cramon, D. Y. (2008). The extended language network: a meta-analysis of neuroimaging studies on text comprehension. *Human Brain Mapping*, 29(5), 581–593.

Ferstl, E. C., & von Cramon, D. Y. (2002). What does the frontomedian cortex contribute to language processing: coherence or theory of mind? *NeuroImage*, 17(3), 1599–1612.

Holyoak, K. J., & Monti, M. M. (2020). Relational integration in the human brain: A review and synthesis. *Journal of Cognitive Neuroscience*, 1–15.

Jacoby, N., & Fedorenko, E. (2018). Discourse-level comprehension engages medial frontal Theory of Mind brain regions even for expository texts. *Language, Cognition and Neuroscience*, 1–17.

Mar, R. A. (2011). The neural bases of social cognition and story comprehension. *Annual Review of Psychology*, 62, 103–134.

Monti, M. M., & Osherson, D. N. (2012). Logic, language and the brain. *Brain Research*, 1428, 33–42.

Monti, M. M., Parsons, L. M., & Osherson, D. N. (2009). The boundaries of language and thought in deductive inference. *Proceedings of the National Academy of Sciences of the United States of America*, 106(30), 12554–12559.

Nieuwenhuis, S., Forstmann, B. U., & Wagenmakers, E.-J. (2011). Erroneous analyses of interactions in neuroscience: a problem of significance. *Nature Neuroscience*, 14(9), 1105–1107.

Paunov, A. M., Blank, I. A., & Fedorenko, E. (2019). Functionally distinct language and Theory of Mind networks are synchronized at rest and during language comprehension. *Journal of Neurophysiology*, 121(4), 1244–1265.

Poldrack, R. A., Baker, C. I., Durnez, J., Gorgolewski, K. J., Matthews, P. M., Munafò, M. R., ... Yarkoni, T. (2017). Scanning the horizon: towards transparent and reproducible neuroimaging research. *Nature Reviews. Neuroscience*, 18(2), 115–126.

Prado, J. (2018). The relationship between deductive reasoning and the syntax of language in Broca’s area: A review of the neuroimaging literature. *L’annee Psychologique*, 118(3), 289–315.

Prado, J., Léone, J., Epinat-Duclos, J., Trouche, E., & Mercier, H. (2020). The neural bases of argumentative reasoning. *Brain and Language*, 208(104827), 104827.

Reverberi, C., Cherubini, P., Rapisarda, A., Rigamonti, E., Caltagirone, C., Frackowiak, R. S. J., ... Paulesu, E. (2007). Neural basis of generation of conclusions in elementary deduction. *NeuroImage*, 38(4), 752–762.

Reviewer #3 (Remarks to the Author):

Xu and colleagues use intersubject correlation (ISC and ISFC) analyses to compare stimulus-locked responses and functional network configuration between narrative and argumentative stimuli. The distinction between discourse types is supported by behavioral ratings of the stimuli. They show that narrative stimuli preferentially engage the default-mode network, whereas argumentative stimuli increase coupling between frontoparietal and language areas. Overall, the paper is well-written and the figures look good. I outline some concerns about the overall framing of the results, suggest a more direct comparison between the conditions of interest, and have a couple questions about the methods.

Major comments:

The psychological argument that complex thoughts boil down into “two natural kinds”—narrative and argumentative thought—doesn’t seem very compelling to me. The authors point to work by James and Bruner (which I like a lot), but is there any more contemporary empirical evidence supporting this distinction? Why should we believe these are “natural kinds,” rather than just different ends of a discourse spectrum, or different regions of a more complex discourse space? For example, it seems that argumentative texts might sometimes intersperse narrative bits for the sake of explication or keeping the reader engaged (and vice versa); would such a hybrid text still be considered argumentative? This coarse-grained dichotomy doesn’t seem to really respect the complexity of actual discourse. (Also makes me wonder whether modern text analysis or topic modeling methods could better capture the finer-grained flow between narrative and argumentative modes over the course of a single text.)

Following on the previous comment, my main concern with the manuscript is whether there could be some less-interesting “third variable”—such as how “engaging” or “interesting” or “personally relevant” the stimulus is—driving the distinction between the two types of stimuli. If I were to look at the ISC results for these four stimuli outside the context of the authors’ narrative-vs-argumentative framing, my interpretation would simply be: “yes, listening to Calvino is much more engaging than listening to Pinker” (at least in my personal experience, and especially for non-linguists!). This is a very difficult problem to “solve,” though, as narrative discourse may be generally more engaging than argumentative discourse in natural contexts. Can the authors address this concern? For example, I think the behavioral analysis of qualities like “concreteness,” “self-projection,” etc is great, but I don’t see any questions about how engaging participants found the stimulus, or how much they enjoyed reading it. I worry that these metrics would also be higher for narrative texts, which complicates the story. (I also wonder if the authors have any other evidence of engagement; e.g. wakefulness, pupil dilation, etc).

A side note to my first comment is that the jump from (a) two distinct psychological constructs for narrative and argumentative thought to (b) two distinct neural substrates for narrative and argumentative thought seems fairly weak. There doesn’t seem to be much work pointing to a brain-based distinction, so it seems like the authors are playing a balancing act of trying to effectively introduce this potential distinction, and, at the same time, pointing to references to support a preexisting distinction. The authors’ approach to framing this potential distinction as an empirical question (i.e. content-dependent hypothesis vs content-independent hypothesis) makes sense to me in lieu of strong preexisting empirical evidence for a brain-based distinction.

It seems that one of the core comparisons here would be between the intact narrative ISCs and the intact argumentative ISCs. Specifically, I would want to see that the variance between the two intact discourse types is greater than the variance among the two exemplar stimuli within each discourse type (i.e. 2 x 2 ANOVA). However, I don’t see this direct comparison anywhere. The authors focus on various contrasts (e.g. intact > scramble), but it would be helpful to see the (pre-contrast) intact ISC maps. This simpler sort of analysis could better set the reader up to interpret the more complex logic of the contrast analysis (which follow the form of “the difference between intact and scrambled versions of narrative stimuli is greater than the difference between intact and scrambled versions of argumentative stimuli”).

Following on the previous comment, a difficult problem with a design like this is convincing readers that this result will generalize to different exemplar stimuli from the narrative and argumentative modes. With this in mind, I think the analysis of consistency across stimulus exemplars within a discourse type could benefit from a more formal treatment. For example, a correlation between the unthresholded intact ISC maps for each exemplar within a discourse type (or ICC?) would provide a more quantitative measure of consistency than just eyeballing the (thresholded) maps in Figure S4 and S5. (And again, in something like Figure S3, S4, or S5 it would be nice to see the pre-contrast intact ISC maps for all four stimuli).

I'm curious why we don't see any real difference in ISC maps between intact and scrambled argumentative stimuli (line 146, 155, Figure 2). I can imagine a scenario where: (a) narrative stimuli require the ordered sequence of events where B -> A -> C to be comprehensible and a sentence-scrambled version like C -> A -> B is nonsensical; while (b) argumentative stimuli are structured such that A -> B -> C and B -> A -> C are effectively identical. Is this the case with these stimuli? Or could this be a result of playing the scrambled stimuli before the intact stimuli during the scanning session? More discussion of why the ISC maps for intact and scrambled argumentative stimuli are so similar would be helpful.

I didn't fully understand the method for mitigating confounds described at lines 468–473 (maybe because I'm not very familiar with ICA-AROMA). As far as I understand, you use ICA-AROMA to obtain both signal and motion components. Then you use a GLM to fit these components to the original BOLD signal, which yields regression coefficients (betas). Then you subtract(?) the betas(?) for the motion components from the... what? I'm confused about the sentence "We estimated the beta coefficients using the fitglm function in Matlab 2019a and subtracted the motion-relevant terms from the BOLD signal." Do you mean you subtract the motion-related IC time series from the BOLD time series? In that case, what's the role of the betas? Is there a precedent in the literature for using this "non-aggressive" approach to confounds, and why would we want to "preserve the shared variance between the motion-relevant components and the signal components" (which could simply be due to motion)? Then you regress out the mean time series for CSF and white matter? I'm confused about the distinction between "subtracting the terms(?) from the signal" and the more usual approach of regressing out confound signals (i.e. fitting a regression equation with confound variables and retaining the residual time series for further analysis). In any case, intersubject correlation analyses are fairly robust to idiosyncratic noise, so I doubt the details of confound mitigation make a huge difference here—but I'd like to better understand what's going on. (I would also be cautious about multi-stage filtering in your approach; Lindquist et al., 2019.)

For contrasting paired ISC values, a permutation test randomly shuffling the group assignments may be a more sensitive (and appropriate) test than using a bootstrap hypothesis test. (Or, equivalently, randomly flipping the signs of the paired contrast across subjects at each permutation; 2^{*16} possible permutations). My reasoning here is that this sort of permutation test would accommodate the paired structure of the samples whereas the bootstrap does not. I'm also a bit unsure about allowing for a left-out subject to be randomly sampled into the average of $N - 1$ subjects in a bootstrap sample (line 525). This might make your bootstrap distribution positively skewed (i.e. higher than the original mean leave-one-out ISC), which would result in an overly stringent null distribution (although I'm not sure it would affect contrasts the same way). Same for the ISFC analysis. Note that in our BrainIAK implementation of the bootstrap hypothesis test for leave-one-out ISC, we resample the ISC values themselves (i.e. after computing leave-one-out ISC) rather than resampling the subjects, then computing ISC (<https://github.com/brainiak/brainiak/blob/master/brainiak/isc.py#L650>).

Minor comments:

When reading the Results, it wasn't obvious to me that subjects were listening to an auditory recording of the stimulus rather than reading a text version of the stimulus (didn't realize this until line 409).

Line 173: I would expand this "SES" acronym when you initially introduce it (and point to

reference 60)

Lines 242, 255, 582: "insignificant" > "nonsignificant"

Line 269: "stimuli-evoked" > "stimulus-evoked"

Line 275: "The argumentative" > "Argumentative"

Line 276: "the functional couplings" > "functional coupling"

Line 285: "after" > "for"

Line 306: "simultaneously featured by two factors" sounds odd to me

Line 326: "yeo" > "Yeo"

Line 327: "unite" > "unit"

Line 359: "that failed to be segmented" > "that segmentation failed"

Line 371: "initialing" > "initial"

Line 548: "The averaged the" > "The averaged"

Line 550: "pair-wised" > "pairwise"

Figure S7: "fsaverag5" > "fsaverage5"

Figure S7: "metric-to-volume" > "metric-to-volume"

I encourage the authors to share their code (e.g. via GitHub). This seems like a very rich naturalistic dataset, and I hope the authors plan to share it publicly (e.g. via OpenNeuro).

References:

Lindquist, M. A., Geuter, S., Wager, T. D., & Caffo, B. S. (2019). Modular preprocessing pipelines can reintroduce artifacts into fMRI data. *Human Brain Mapping, 40*(8), 2358–2376. <https://doi.org/10.1002/hbm.24528>

Samuel A. Nastase

Reviewer #1 (Remarks to the Author):

Xu et al. used fMRI to examine commonalities and differences in functional activation and functional connectivity patterns during two different forms of thought: argumentative and narrative. The authors took advantage of inter-subject synchronization-based analyses to quantify these patterns. While both modes of thought engaged auditory regions along the superior temporal sulcus and inferior frontal gyrus, narrative thought additionally activated and synchronized default mode network regions. In contrast argumentative thought demonstrated increased synchronization of auditory and control networks. These findings point to somewhat overlapping yet distinct brain activation patterns for these modes of thought. Overall, I really like this study topic and the methods that were used. Understanding the neural correlates of different modes of thought is a key topic to address and has not yet been fully investigated. In this regard the present study will make a nice contribution to the literature as it highlights some potentially interesting differences involving the default mode network. I do have some comments that I hope will strengthen the manuscript.

R: We thank the reviewer for the general positive comments.

1) I think it would be useful to have a little more detail in the definition of the two modes of thought in the first paragraph of the Intro and perhaps include a short example of each. I found myself not completely clear on the core elements that distinguish them. It might also be useful to give some context that will anticipate the questions that were asked of subjects related to abstractness, degree of scene construction, theory of mind etc. To what extent are these defining features of each mode of thought versus details that differed in the specific examples used but may not necessarily distinguish these modes of thought in general. The authors rightly point out that the literature has mainly focused on narrative thought, so the rest of the paper really depends on having readers understand what argumentative thought is and how it differs from narrative thought.

R: We thank the reviewer for raising this issue. In the revised Introduction, we have added two paragraphs and a figure to elaborate on the commonalities and differences between narrative and argumentative thought (the second and third paragraphs in the Introduction, Line 30-55; Figure 1). In the two paragraphs, we introduced the framework of four traditional discourse modes in text linguistics. Based on this framework, we illustrated the core commonalities and differences between narrative and argumentative thought. We further pointed out the distinctive cognitive faculties associated with the two modes of thought (e.g., situation model vs. informal logic), from which our rating items were derived. We hope the additional information can help readers better understand the relation between two thought modes.

2) I thought that the framing of the study was somewhat problematic. The general idea is that the study would use brain data to adjudicate between two hypotheses—that narrative and argumentative thought are fundamentally different or fundamentally the same. However, in order to test this, the authors used stimuli that were chosen specifically because they have different properties, at least from a psychological perspective. So, there is really no information that is gained from the brain in terms of quantifying how different these modes of thought are. Moreover, this framing relies on reverse inference which is a notoriously challenging approach to do correctly (and would require much more justification than is currently offered). I think that the study would be just as strong and not suffer from these issues if it were simply presented for what it is—an investigation of the neural correlates of different modes of thought. This becomes a forward inference approach which has fewer thorny issues to contend with. I don't think there is any problem in mentioning that there are different conceptions of how similar/distinct these modes of thought are. But I do see a problem in framing the study as a test of these hypotheses using brain data.

R: we thank the reviewer for the thoughtful comments. We have revised the introduction (the sixth paragraph of the introduction, Line 77-92). In the revised version, we raised the content-independent and content-dependent

hypotheses purely from the neural substrate's perspective, i.e., whether narrative and argumentative thought engage the same neural substrate or not, without reverse inferring whether the two thought modes are different or not in our mind.

3) The results suggest that the DMN is really only involved in narrative thought and not argumentative thought. This is potentially a very interesting finding. However, one possibility is that the DMN is recruited during argumentative thought, but in a way that is more idiosyncratic across subjects. This possibility could be addressed by presenting results from a standard group level analysis (rather than ISC) using an intact argumentative > scrambled argumentative contrast. If this reveals DMN activation it would suggest that this mode of thought does rely on the DMN but that the precise timing of activation is not consistent across participants (hence no inter-subject synchronization). I think this would provide helpful information that could strengthen (or weaken) the claim that the DMN is specific to narrative thought.

R: We thank the reviewer again for the thoughtful comments. Unfortunately, as the study was dedicated to ISC/ISFC paradigm, we found it impossible to implement the univariate analysis on the current data structure. One of the major issues was that each condition (e.g., *Intact Argument* or *Scrambled Argument*) occupied the whole session/run; there was no common baseline across sessions/runs. It is thus not legitimate to contrast the activation level across sessions/runs. We have tried to use each session's first 10s' music as the common baseline. The results were not interpretable for both modes of thoughts (some weak activation on the visual cortex). It might be due to the music's time being too short (10s music vs. about 8 min text) or the signal at the beginning being not stable.

We also think it is very unlikely that the absence of DMN in the argumentative thought in the ISC results was due to the idiosyncrasy across subjects. We indeed found significant ISFC results for the argumentative thought. The across-subject correlations among the frontoparietal regions and between the IPS and the perisylvian regions were significantly higher in the intact-argumentative condition than those in the scrambled-argumentative condition. Thus, the brain areas were synchronized across subjects during the argumentative thought. In the revised manuscript, we have added Figure 6 to emphasize this data point.

4) To compare functional coupling between the two modes of thought, the authors rely on a set of criteria which include a significant contrast (e.g., intact narrative > scrambled narrative) and a non-significant contrast (e.g., intact argumentative > scrambled argumentative). However, this is generally not sufficient to demonstrate a condition difference. A direct contrast (e.g., intact narrative > intact argumentative) is necessary. Or it could be in the form of [intact narrative > scrambled narrative] > [intact argumentative > scrambled argumentative]. Without a direct contrast it is not possible to make conclusions about the specificity of various coupling patterns.

R: Following the reviewer's suggestion, we have validated both ISC and ISFC results based on a direct comparison between narrative thought and argumentative thought. The neural correlates specific of narrative thought were defined as areas or functional couplings that met the following three criteria simultaneously: (1) (*Intact Narrative - Scrambled Narrative*) > (*Intact Argument - Scrambled Argument*); (2) *Intact Narrative* > *Scrambled Narrative*; (3) *Intact Narrative* > *Resting State*. The neural correlates specific of argumentative thought were defined as areas or functional couplings that met the following three criteria simultaneously: (1) (*Intact Argument - Scrambled Argument*) > (*Intact Narrative - Scrambled Narrative*); (2) *Intact Argument* > *Scrambled Argument*; (3) *Intact Argument* > *Resting State*. The ISC results were shown in Line 178-184 and Supplementary Figure 4. The ISFC results were shown in Line 291-297 and Supplementary Figure 11 for narrative thought, in Line 313-320 and Supplementary Figure 12 for argumentative thought. The results defined based on a direct comparison between narrative and argumentative thought were highly similar to those defined initially.

A direct comparison between narrative and argumentative thought indeed statistically justifies our conclusion regarding the "specificity" of one particular type of thought. However, this definition also carried the risk of

involving the neural correlates engaged in both thoughts (*Intact Narrative > Scrambled Narrative* && *Intact Argument > Scrambled Argument*) but show significantly greater ISC/ISFC values during one of them. Considering these pros and cons, we decided to show readers the results using both definitions. One was in the manuscript; one was in the supplementary material.

5) *I found Figure 3 a bit hard to interpret. The purple color bar was not very intuitive and personally, I find circular embedded graph layouts less intuitive than spring embedded graph layouts. However, I realize that these are just my subjective opinions and I completely leave it to the authors to consider alternative formats or to retain their initial choices.*

R: Considering the reviewer's suggestions, we spent some time tuning the color of the bars in the figure. As we could not find a solution to significantly improve the illustration, we decided to keep the original figures.

This study used the force-directed graph drawing algorithm (Fruchterman & Reingold, 1991) to draw the network layout. This algorithm should belong to the spring embedded layout (mentioned by the reviewer), where nodes with weak connections are pushed apart. To avoid this misunderstanding, we have clarified in the manuscript (Line 681-685): “The strength of the repulsive force between nodes was set as 30. This parameter provided a better view of the internal structure among the core regions in the center, e.g., whether the control and the language systems were connected during the argumentative conditions. The byproduct is that the peripheral and less important nodes would have not enough room to be fully extended and be extruded into a circle.” We could decrease the repulsive force between nodes. However, it would make the internal structures among the core regions collapse.

Fruchterman, T. M., & Reingold, E. M. (1991). Graph drawing by force-directed placement. *Software: Practice and experience*, 21(11), 1129-1164.

6) *Line 48: “Another study further illustrates that such higher synchronization in the DMN not only exist between...” should read “Another study further illustrated that such higher synchronization in the DMN not only exists between...”*

R: We have revised the sentence as the reviewer suggested (Line 73).

7) *On line 173, SES is not defined. I also found myself desiring more commonly used statistics in the text and figures such as t and r values. Is there some reason why these could not be included as well?*

R: We thank the reviewer for pointing out this error. We have added the annotation and the reference where SES first appeared (Line 214-216).

We used SES as statistical index because it was suitable to our current analysis method. In this study, we used the inter-subject measures, i.e., ISC and ISFC. We correlated each participant's BOLD signal and the mean BOLD signal of the rest of them. Thus, although each participant has a correlation coefficient, they were not independent. We could not make statistical inferences across participants by implementing a traditional t-test. As the BOLD signal is autocorrelated, the time points are not independent, either. We thus could also not make statistical inferences across time points by implementing a traditional r test. We adopted the subject-wise bootstrapping method to make inferences, a method dedicated to the inter-subject approach, where both exchangeability and independence assumptions are satisfied (Chen et al., 2016). We generated the null distribution with this method and defined the standardized effect size based on the null distribution: $(x - \mu)/\sigma$, where x is the veritable contrast value, μ is the mean of the null distribution, and σ is the standard deviation of the null distribution (Botta-Dukát, 2018).

Chen, G., Shin, Y. W., Taylor, P. A., Glen, D. R., Reynolds, R. C., Israel, R. B., & Cox, R. W. (2016). Untangling the relatedness among correlations, part I: nonparametric approaches to inter-subject correlation analysis at the group level. *NeuroImage*, 142, 248-259.

Botta-Dukát, Z. (2018). Cautionary note on calculating standardized effect size (SES) in randomization test. *Community Ecology*, 19(1), 77-83.

8) *On line 239, I'm not familiar with the term "temporal entrance".*

R: We have added the annotation and the reference for this brain area (Line 276).

9) *Line 310, I believe "medial" is meant to be "middle"*

R: We thank the reviewer for pointing out this typo. We have revised this word as the reviewer suggested (Line 378).

10) *Line 327, "unite" should be "unit"*

R: We have revised the word as the reviewer suggested (Line 393).

Reviewer #2 (Remarks to the Author):

In this manuscript, Xu and colleagues use the method of intersubject correlations (ISC) to examine the fMRI correlates of narrative and argumentative thoughts in adults. They found commonalities and differences in activity and connectivity between both types of discourses.

I think that the study addresses a very interesting question, using a solid methodological approach that may shed light on the brain mechanisms underlying discourse processing in different contexts. However, the manuscript also has several weaknesses and would benefit from a revision to strengthen its conclusions and clarify its contribution to the current literature. My comments are listed below.

R: We thank the reviewer for the general positive comments.

Major issues:

1) The literature review is sparse. There is quite an extensive literature on the brain mechanisms underlying discourse processing and their relation with the default mode / Theory of Mind network (e.g., Ferstl, Neumann, Bogler, & von Cramon, 2008; Ferstl & von Cramon, 2002; Jacoby & Fedorenko, 2018; Mar, 2011; Paunov, Blank, & Fedorenko, 2019; Prado, Léone, Epinat-Duclos, Trouche, & Mercier, 2020). Although some of these studies are briefly mentioned, many are missing and there is very little discussion of this literature in either the introduction of the discussion. This is quite surprising as several of these studies have examined questions that are related to the present investigation. For example, back in 2002, Ferstl and von Cramon already looked at the neural correlates of logical inferences during discourse comprehension. I believe that the paper would benefit from a more thorough discussion of this literature, which would also clarify the contribution of the present paper with respect to these previous studies.

Similarly, the authors make assumptions about what could be expected in terms of neural bases of argumentative thought based on a very limited literature. For example, one reference to “informal logic” from a specific chapter led them to infer that, according to the content-dependent hypothesis, argumentative discourse “might warrant cooperation between the language and the reasoning brain systems”. There is an extensive field in the psychology of reasoning whose goal is to determine whether logical reasoning related to natural language (e.g., see Monti & Osherson, 2012; Prado, 2018). There is also an even more relevant literature on the neuroimaging of reasoning investigating the very question of whether logical thought is supported by the mechanism of language (e.g., Holyoak & Monti, 2020; Monti, Parsons, & Osherson, 2009; Reverberi et al., 2007). I think the paper would benefit from a discussion of this literature, again so that the contribution of the present manuscript can be clearly fleshed out.

R: We thank the reviewer very much for the recommendation of these relevant studies. We have incorporated these relevant studies in the revised manuscript. In the discussion, we have added one paragraph to discuss why our results seem to contradict the findings by Ferstl and von Cramon (2002), in which the results are usually interpreted as evidence supporting the DMN is the content-independent network for discourse processing (Line 362-371). In the discussion, we also added the literature discussion to support why cooperation between language and frontoparietal control systems might be crucial to the informal logic process (Line 354-361). We hope the manuscript is improved by adding these discussions.

2) There is no direct comparison of narrative versus argumentative texts. The authors attempt to show that there are differences in the neural bases of narrative and argumentative texts by combining maps showing significant effects for one condition and insignificant effects in the other. Yet, this does not show that the effect in one condition is greater than the effect in the other. This would need to be statistically demonstrated, for example by directly

comparing the contrast Intact Argument > Scrambled Argument to the contrast Intact Narrative > Scrambled Narrative (and vice versa). See Nieuwenhuis, Forstmann, & Wagenmakers (2011) for a discussion of this classic issue in neuroimaging studies.

R: We thank the reviewer for the comments. The first reviewer raised the same issue (see the fourth comment). We copy the answer below.

Following the reviewer's suggestion, we have validated both ISC and ISFC results based on a direct comparison between narrative thought and argumentative thought. The neural correlates specific of narrative thought were defined as areas or functional couplings that met the following three criteria simultaneously: (1) *(Intact Narrative - Scrambled Narrative) > (Intact Argument - Scrambled Argument)*; (2) *Intact Narrative > Scrambled Narrative*; (3) *Intact Narrative > Resting State*. The neural correlates specific of argumentative thought were defined as areas or functional couplings that met the following three criteria simultaneously: (1) *(Intact Argument - Scrambled Argument) > (Intact Narrative - Scrambled Narrative)*; (2) *Intact Argument > Scrambled Argument*; (3) *Intact Argument > Resting State*. The ISC results were shown from Line 178-184 and Supplementary Figure 4. The ISFC results were shown in Line 291-297 and Supplementary Figure 11 for narrative thought, in Line 313-320 and Supplementary Figure 12 for argumentative thought. The results defined based on a direct comparison between narrative and argumentative thought were highly similar to those defined initially.

A direct comparison between narrative and argumentative thought indeed statistically justifies our conclusion regarding the "specificity" of one particular type of thought. However, this definition also carried the risk of involving the neural correlates engaged in both thoughts (*Intact Narrative > Scrambled Narrative && Intact Argument > Scrambled Argument*) but show significantly greater ISC/ISFC values during one of them. Considering these pros and cons, we decided to show readers the results using both definitions. One was in the manuscript; one was in the supplementary material.

3) The resting state data could be used to identify the DMN in each subject. At the heart of the analysis strategy is the idea to compare the brain networks subserving narrative and argumentative thought to the DMN. However, although the authors do acquire some resting state data, they only make use of these data as a control condition against which maps associated with texts are compared. Instead of comparing activity/connectivity associated with texts to general regions of the DMN defined by a brain atlas, I think it would strengthen the authors' conclusions to (a) visualize the DMN for this specific group of participants and (b) use these specific regions to compare the DMN to the network associated with the contrasts of intact versus scrambled texts. This would give more specificity to the inferences and increase confidence in the conclusions regarding overlap with the DMN.

R: Following the reviewer's suggestion, we have defined the DMN using the data specifically from the participants in this experiment. The DMN was mapped using the see-based resting-state functional connectivity from the posterior cingulate cortex, one of the core regions in the DMN (Line 150-154; see Line 568-577 for the method details). In the revised manuscript, we have (a) visualized the DMN for this specific group in Supplementary Figure 2b and (b) updated Figure 3, Supplementary Figure 6, and Supplementary Figure 7 to compare the DMN to the significant areas in the contrasts between the intact and the scrambled conditions. This newly defined DMN closely resembled the one defined by the brain atlas and did not change our conclusion: narrative thought engages brain areas in the DMN, but argumentative thought does not.

4) The discussion relies too heavily on reverse inferences. I think the authors should keep in mind that they do not directly localize the brain regions supporting language processing, attentional processing, visual processing or even the DMN. These are simply taken from an atlas. Therefore, they should be careful not to overinterpret involvement of the different brain regions as a function of these processes. For example, on the basis on one single study, they claim that the IPS is the "neural basis of fluid intelligence" (line 302) and therefore argue that "the

coordination and cooperation between the frontoparietal control system and the language system, which is mediated by the IPS, might be critical to identify and evaluate the informal logic in the natural language discourse.” This is extremely speculative as one could find a long list of functions subserved by the IPS besides fluid intelligence (e.g., attention, working memory, quantitative processing etc.). The authors should stay away from these sorts of inferences (there are others in the discussion, for example regarding the role of regions in the DMN, the frontoparietal system etc.) and focus on what their study show in terms of commonalities and quantitative differences between narrative and argumentative texts. For this, a direct comparison of conditions is necessary (see point #2 above).

R: We thank the reviewer for raising this issue. We have removed the discussion about the function of IPS. Since we still lack a stable consensus about individual regions' functions, we also avoided discussing individual regions' functions throughout the paper. In the discussion, we still provide the necessary explanation of each result based on our hypotheses, incorporating the general view about each brain system's function in the previous literature. We have implemented a direct comparison between conditions (see the response to point 2).

5) There are inconsistencies in the hypotheses. The authors argue in the introduction that two hypotheses can be made regarding argumentative and narrative thought. On the one hand, the content-dependent hypothesis posits that the DMN should only support narrative texts and not argumentative texts. On the other hand, the content-independent hypothesis posits that both texts should be supported by the DMN, which “might potentially serve as general machinery for long-timescale information integration, supporting both narrative and argumentative thought.” (line 73). If I understand correctly this framework as it is laid out in the introduction, the results clearly give support to the content-dependent hypothesis as the DMN is only found in narrative texts. However, the authors appear to slightly change the hypothesis in the discussion, arguing that because there is overlap in other regions than the DMN (notably the frontoparietal network), the content-independent hypothesis is still supported (e.g., “The results revealed the commonalities and differences between neural bases underlying narrative and argumentative thought, which seems to support both content-independent and content-dependent hypotheses”). This seems inconsistent with what is stated in the introduction to me. If the theory makes the prediction that the DMN is the center of this content-independent processing, then I do not see how it is supported. The authors should clarify the hypotheses, make sure they are consistent throughout the paper and interpret their results with respect to the hypothesized framework.

R: We are sympathetic to the reviewer's confusion. In the revised manuscript, we have emphasized that the hypotheses were "partially" supported (Line 332). When discussing the content-independent hypothesis, we also acknowledge the reader "However, instead of the DMN, as the hypothesis initially predicted in the Introduction, we found the shared neural basis for both narrative and argumentative thought in the frontoparietal control system." (Line 335-337). We hope the revision can improve the consistency of the manuscript.

6) The sample size is rather small. After excluding several participants, only 16 participants are analyzed. This seems quite small and the study might be underpowered if one considers typical effect sizes in neuroimaging studies (Poldrack et al., 2017). The only rationale given is that “the sample size was in line with the studies employing ISC and ISFC methods (11 and 18 participants, respectively).” This is not very convincing, as it is well known that most prior neuroimaging studies are largely underpowered (Button et al., 2013) and it is likely that both of these studies are also underpowered. The authors should provide a clear justification for their final sample size, ideally based on a power analysis.

R: Following the reviewer's and the editor's suggestion, we calculated the observed power for both ISC and ISFC analyses under our sample size (see below for details). The results show that the significant ISC and ISFC results reported in the manuscript have adequate power to be detected. As both narrative and argumentative results were obtained from the same group of participants, the nonsignificant result for the argumentative thought in the ISC

results should result from the trivial effect size rather than from the small sample size. We hope this power analysis can justify our sample size.

We used the same bootstrapping method to implement power analysis (see Method for details). Both the null distribution and the alternative distribution (i.e., the distribution when H1 is true) were assumed to follow the same distribution as the sampling distribution. We generated the null distribution by subtracting the veritable contrast value from the sample distribution. The alternative distribution, estimated as the distribution surrounding the veritable effect size, was the same as the sampling distribution. We calculated the power as the percentages of the alternative distributions that were greater than the threshold.

The following figure illustrates each vertical's power in the ISC results. In the contrasts *Intact Narrative > Scrambled Narrative*, and *Narrative Thought (Intact Narrative - Scrambled Narrative) > Argumentative Thought (Intact Argument - Scrambled Argument)*, we set the threshold as FDR corrected $P < 0.05$ (same as the one reported in the manuscript). Since there were no significant areas in the contrasts *Intact Argument > Scrambled Argument*, and *Argumentative Thought (Intact Narrative - Scrambled Narrative) > Narrative Thought (Intact Argument - Scrambled Argument)*, we set the threshold as uncorrected $P < 0.001$. We found that significant regions in the narrative results did have enough power to be detected (left column of the figure). However, no areas in the argumentative results had enough power even under a lower threshold as uncorrected $P < 0.001$ (right column of the figure). As both narrative and argumentative results were obtained from the same group of participants, the nonsignificant results in the argumentative results should result from the trivial effect size, not from the small sample size.

We further calculated each functional coupling's power in the ISFC results. The threshold was set as FWE corrected $P < 0.05$ (same as the one reported in the manuscript). For both narrative and argumentative results, most of the significant functional couplings had more than 80% power to be detected (see the table below).

Contrasts	Percentages of Significant ISFC over 80% Power
Intact Narrative > Scrambled Narrative	82.60%
Intact Argument > Scrambled Argument	75.58%
Narrative Thought > Argumentative Thought	80.55%
Argumentative Thought > Narrative Thought	74.39%

Minor issues:

7) *If I understand correctly, the authors use ratings from another group of participants to classify texts as narrative versus argumentative. It is then not surprising that, in this specific group, the ratings differ significantly with respect to the measures that were used to classify the texts. The results reported from line 100 to 107 are thus a bit trivial and this should be more clearly indicated. Additionally, what is less trivial is whether the participants in the fMRI experiment also differed with respect to the rating. These results are presented in Figure S1, I would move them to the main text (and also move Figure S1 to the manuscript).*

R: Following the reviewer's suggestion, we have switched the two figures and added the statistical information for Figure 2 (Line 128-134).

8) *line 506: "We calculated the ISC for each vertex each run" should be "We calculated the ISC for each vertex and each run"*

R: We thank the reviewer for pointing out this typo. We have revised the sentence as the reviewer suggested (Line 581).

Button, K. S., Ioannidis, J. P. A., Mokrysz, C., Nosek, B. A., Flint, J., Robinson, E. S. J., & Munafò, M. R. (2013). Power failure: why small sample size undermines the reliability of neuroscience. Nature Reviews. Neuroscience, 14(5), 365–376.

Ferstl, E. C., Neumann, J., Bogler, C., & von Cramon, D. Y. (2008). The extended language network: a meta-analysis of neuroimaging studies on text comprehension. Human Brain Mapping, 29(5), 581–593.

Ferstl, E. C., & von Cramon, D. Y. (2002). What does the frontomedian cortex contribute to language processing: coherence or theory of mind? NeuroImage, 17(3), 1599–1612.

Holyoak, K. J., & Monti, M. M. (2020). Relational integration in the human brain: A review and synthesis. Journal of Cognitive Neuroscience, 1–15.

Jacoby, N., & Fedorenko, E. (2018). Discourse-level comprehension engages medial frontal Theory of Mind brain regions even for expository texts. Language, Cognition and Neuroscience, 1–17.

Mar, R. A. (2011). The neural bases of social cognition and story comprehension. Annual Review of Psychology, 62, 103–134.

Monti, M. M., & Osherson, D. N. (2012). Logic, language and the brain. Brain Research, 1428, 33–42.

Monti, M. M., Parsons, L. M., & Osherson, D. N. (2009). The boundaries of language and thought in deductive inference. Proceedings of the National Academy of Sciences of the United States of America, 106(30), 12554–12559.

Nieuwenhuis, S., Forstmann, B. U., & Wagenmakers, E.-J. (2011). Erroneous analyses of interactions in neuroscience: a problem of significance. Nature Neuroscience, 14(9), 1105–1107.

- Paunov, A. M., Blank, I. A., & Fedorenko, E. (2019). Functionally distinct language and Theory of Mind networks are synchronized at rest and during language comprehension. *Journal of Neurophysiology*, 121(4), 1244–1265.
- Poldrack, R. A., Baker, C. I., Durnez, J., Gorgolewski, K. J., Matthews, P. M., Munafò, M. R., ... Yarkoni, T. (2017). Scanning the horizon: towards transparent and reproducible neuroimaging research. *Nature Reviews Neuroscience*, 18(2), 115–126.
- Prado, J. (2018). The relationship between deductive reasoning and the syntax of language in Broca's area: A review of the neuroimaging literature. *L'annee Psychologique*, 118(3), 289–315.
- Prado, J., Léone, J., Epinat-Duclos, J., Trouche, E., & Mercier, H. (2020). The neural bases of argumentative reasoning. *Brain and Language*, 208(104827), 104827.
- Reverberi, C., Cherubini, P., Rapisarda, A., Rigamonti, E., Caltagirone, C., Frackowiak, R. S. J., ... Paulesu, E. (2007). Neural basis of generation of conclusions in elementary deduction. *NeuroImage*, 38(4), 752–762.

Reviewer #3 (Remarks to the Author):

Xu and colleagues use intersubject correlation (ISC and ISFC) analyses to compare stimulus-locked responses and functional network configuration between narrative and argumentative stimuli. The distinction between discourse types is supported by behavioral ratings of the stimuli. They show that narrative stimuli preferentially engage the default-mode network, whereas argumentative stimuli increase coupling between frontoparietal and language areas. Overall, the paper is well-written and the figures look good. I outline some concerns about the overall framing of the results, suggest a more direct comparison between the conditions of interest, and have a couple questions about the methods.

R: We thank the reviewer for the general positive comments.

Major comments:

The psychological argument that complex thoughts boil down into “two natural kinds”—narrative and argumentative thought—doesn’t seem very compelling to me. The authors point to work by James and Bruner (which I like a lot), but is there any more contemporary empirical evidence supporting this distinction? Why should we believe these are “natural kinds,” rather than just different ends of a discourse spectrum, or different regions of a more complex discourse space? For example, it seems that argumentative texts might sometimes intersperse narrative bits for the sake of explication or keeping the reader engaged (and vice versa); would such a hybrid text still be considered argumentative? This coarse-grained dichotomy doesn’t seem to really respect the complexity of actual discourse. (Also makes me wonder whether modern text analysis or topic modeling methods could better capture the finer-grained flow between narrative and argumentative modes over the course of a single text.)

R: We thank the reviewer for the thoughtful comments. In the revised introduction, we introduced the framework of macrogenres in text linguistics and the empirical evidence supporting this division. From this framework, we further pointed out the core commonalities and differences between narrative and argumentative thought (the second and the third paragraph of the introduction, Line 30-55; Figure 1). We hope the added information can help to justify the validity of thought dichotomy.

The reviewer also raised an interesting issue about hybrid thought. On the one hand, we do believe in the existence of hybrid thought. The current study, which adopted relatively pure narrative and argumentative texts, can be considered as a start to investigate more complicated thoughts. On the other hand, we also believe the concrete (narrative) vs. abstract (argumentative) dichotomy might still represent the basic modes of human thinking. Evidence has suggested that genre expectations significantly affect how readers process and remember texts (Zwaan, 1991, 1993, 1994). For instance, when a story is read alone, the reader is most likely to treat it as a literary work, focusing on the plots and characters' intentions. However, when the same story is inserted into an argumentative text as an example to support a proposition, the reader may pay more attention to the story's general implication and evaluate whether it can justify the conclusion. Further neuroimaging studies should address this issue.

Zwaan, R. A. (1991). Some parameters of literary and news comprehension: Effects of discourse-type perspective on reading rate and surface structure representation. *Poetics*, 20(2), 139-156.

Zwaan, R. A. (1992). Aspects of literary comprehension: A cognitive approach (Doctoral dissertation, Rijksuniversiteit te Utrecht).

Zwaan, R. A. (1994). Effect of genre expectations on text comprehension. *Journal of experimental psychology: learning, memory, and cognition*, 20(4), 920.

Following on the previous comment, my main concern with the manuscript is whether there could be some less-interesting “third variable”—such as how “engaging” or “interesting” or “personally relevant” the stimulus is—driving the distinction between the two types of stimuli. If I were to look at the ISC results for these four stimuli outside the context of the authors’ narrative-vs-argumentative framing, my interpretation would simply be: “yes, listening to Calvino is much more engaging than listening to Pinker” (at least in my personal experience, and especially for non-linguists!). This is a very difficult problem to “solve,” though, as narrative discourse may be generally more engaging than argumentative discourse in natural contexts. Can the authors address this concern? For example, I think the behavioral analysis of qualities like “concreteness,” “self-projection,” etc is great, but I don’t see any questions about how engaging participants found the stimulus, or how much they enjoyed reading it. I worry that these metrics would also be higher for narrative texts, which complicates the story. (I also wonder if the authors have any other evidence of engagement; e.g. wakefulness, pupil dilation, etc).

R: We thank the reviewer again for the thoughtful comments. Mar et al. (2021) did a meta-analysis including 75 unique samples and data from more than 33,000 participants and found that stories are more easily understood and better recalled than essays, even when the content was matched. This difference thus should owe to the nature of the two text types (Mar et al., 2021). For instance, stories are experience-based, essays are remote from experience. To understand a narrative, we need to simulate the state of affairs and self-project into the situation. In this sense, we do not think engagement is the factor that we should/can match across text types.

We share the same concern with the reviewer. Throughout the study, whether implementing the ISC or the ISFC analysis, we always first conducted simple contrast within the text type (e.g., *Intact Argument* vs. *Scrambled Argument*) before comparing across text types. Thus, the absence of the DMN should not result from a lower level of engagement in the argumentative condition than the narrative one. The intact argumentative texts (5 point rating: 4.41 ± 0.74) were much more comprehensible than the scrambled argumentative text (5 point rating: 2.97 ± 0.99) (pair $t(15) = 8.46$, $P < 0.001$).

More importantly, in the ISFC results, we did find the significant functional couplings specific for argumentative thought. Such a double disassociation pattern cannot be explained by one single factor like engagement.

Mar, R. A., Li, J., Nguyen, A. T., & Ta, C. P. (2021). Memory and comprehension of narrative versus expository texts: A meta-analysis. *Psychonomic Bulletin & Review*, 1-18.

A side note to my first comment is that the jump from (a) two distinct psychological constructs for narrative and argumentative thought to (b) two distinct neural substrates for narrative and argumentative thought seems fairly weak. There doesn’t seem to be much work pointing to a brain-based distinction, so it seems like the authors are playing a balancing act of trying to effectively introduce this potential distinction, and, at the same time, pointing to references to support a preexisting distinction. The authors’ approach to framing this potential distinction as an empirical question (i.e. content-dependent hypothesis vs content-independent hypothesis) makes sense to me in lieu of strong preexisting empirical evidence for a brain-based distinction.

R: we thank the reviewer's agreement. The first reviewer raised the same issue (see the second point). The current study indeed aims to investigate what is the neural substrate underlying narrative and argumentative thought and whether they share the same neural substrate or not. We further clarified this point in the sixth paragraph

It seems that one of the core comparisons here would be between the intact narrative ISCs and the intact argumentative ISCs. Specifically, I would want to see that the variance between the two intact discourse types is greater than the variance among the two exemplar stimuli within each discourse type (i.e., 2 x 2 ANOVA). However, I don’t see this direct comparison anywhere. The authors focus on various contrasts (e.g. intact > scramble), but it would be helpful to see the (pre-contrast) intact ISC maps. This simpler sort of analysis could better set the reader

up to interpret the more complex logic of the contrast analysis (which follow the form of “the difference between intact and scrambled versions of narrative stimuli is greater than the difference between intact and scrambled versions of argumentative stimuli”).

R: Following the reviewer's suggestion, we provided the visualizations of the pre-contrast raw ISC maps of the four intact-text conditions (Supplementary Figure 5a). As "exemplar within each discourse type" is a random variable, not a fixed variable, we cannot set levels to this variable and implement a 2 by 2 ANOVA. To answer whether the variance between the two discourse types was greater than the variance within each discourse type, we followed the suggestion from the reviewer's next comment. We implemented the Pearson's correlation among the raw ISC maps of four intact-text conditions across vertices. We indeed found texts of the same types induced more similar ISC patterns than the texts of different types (Supplementary Figure 5b).

Following on the previous comment, a difficult problem with a design like this is convincing readers that this result will generalize to different exemplar stimuli from the narrative and argumentative modes. With this in mind, I think the analysis of consistency across stimulus exemplars within a discourse type could benefit from a more formal treatment. For example, a correlation between the unthresholded intact ISC maps for each exemplar within a discourse type (or ICC?) would provide a more quantitative measure of consistency than just eyeballing the (thresholded) maps in Figure S4 and S5. (And again, in something like Figure S3, S4, or S5 it would be nice to see the pre-contrast intact ISC maps for all four stimuli). I'm curious why we don't see any real difference in ISC maps between intact and scrambled argumentative stimuli (line 146, 155, Figure 2). I can imagine a scenario where: (a) narrative stimuli require the ordered sequence of events where B -> A -> C to be comprehensible and a sentence-scrambled version like C -> A -> B is nonsensical; while (b) argumentative stimuli are structured such that A -> B -> C and B -> A -> C are effectively identical. Is this the case with these stimuli? Or could this be a result of playing the scrambled stimuli before the intact stimuli during the scanning session? More discussion of why the ISC maps for intact and scrambled argumentative stimuli are so similar would be helpful.

R: Following the reviewer's suggestion, we provided the visualizations of the pre-contrast raw ISC maps of the four intact-text conditions (Supplementary Figure 5a). We also quantitatively measure the consistency among the intact-text conditions by implementing the Pearson's correlation among the raw ISC maps across vertices. The texts of the same types induced more similar ISC patterns than the texts of different types (Supplementary Figure 5b).

In our stimuli, intact texts were much more comprehensible than the scrambled texts, no matter the texts were narrative or argumentative (Line 135-140): The comprehensibility rating on the intact narrative texts (mean \pm SD: 4.69 ± 0.51) was significantly higher than the scrambled narrative texts (mean \pm SD: 2.63 ± 0.67) (pair $t(15) = 15.17$, $P < 0.001$), and the comprehensibility rating on the argumentative texts (mean \pm SD: 4.41 ± 0.74) was significantly higher than the scrambled argumentative texts (mean \pm SD: 2.97 ± 0.99) (pair $t(15) = 8.46$, $P < 0.001$).

We understand the reviewer's curiosity about the lack of significant areas for argumentative thought (*Intact Argument > Scrambled Argument*) in the ISC analyses. However, we did find neural substrate for argumentative thought in the ISFC analysis, namely increased functional couplings within the control and language systems. This result suggests that, compared to the scrambled-argumentative condition, the intact-argumentative condition did not involve additional brain systems (thus no significant regions in the ISC analyses) but enhanced the frontoparietal functional couplings within the control system and the functional couplings between the IPS in the control system and multiple perisylvian areas in the language system.

In other words, we think that in the light of the ISFC results, the fact that the ISC maps for intact and scrambled argumentative stimuli were so similar is not as surprising as if the ISC results were seen in isolation. To emphasize this data point, we have added Figure 6 in the manuscript, which shows the overlap between the ISC results (*Intact Argument > Resting State*) and ISFC results (*Intact Argument > Scrambled Argument*). The figure shows that the

intact-argumentative condition increased the functional couplings between the regions that showed comparable ISC values to the scrambled argumentative condition.

I didn't fully understand the method for mitigating confounds described at lines 468–473 (maybe because I'm not very familiar with ICA-AROMA). As far as I understand, you use ICA-AROMA to obtain both signal and motion components. Then you use a GLM to fit these components to the original BOLD signal, which yields regression coefficients (betas). Then you subtract(?) the betas(?) for the motion components from the... what? I'm confused about the sentence "We estimated the beta coefficients using the fitglm function in Matlab 2019a and subtracted the motion-relevant terms from the BOLD signal." Do you mean you subtract the motion-related IC time series from the BOLD time series? In that case, what's the role of the betas? Is there a precedent in the literature for using this "non-aggressive" approach to confounds, and why would we want to "preserve the shared variance between the motion-relevant components and the signal components" (which could simply be due to motion)? Then you regress out the mean time series for CSF and white matter? I'm confused about the distinction between "subtracting the terms(?) from the signal" and the more usual approach of regressing out confound signals (i.e. fitting a regression equation with confound variables and retaining the residual time series for further analysis). In any case, intersubject correlation analyses are fairly robust to idiosyncratic noise, so I doubt the details of confound mitigation make a huge difference here—but I'd like to better understand what's going on. (I would also be cautious about multi-stage filtering in your approach; Lindquist et al., 2019.)

R: We thank the reviewer for raising these doubts. We have revised the paragraph about the method we used to exclude the non-neuronal source noise (Line 535-537, Line 543-545). We hope the revision can clear up the misunderstanding. Below is a more detailed response.

The inter-subject correlation method indeed can filter out non-neuronal correlations (Simony et al., 2016). However, noises induced by non-neuronal sources, e.g., heartbeat, respiration, and head motion, may blur the stimulus-induced signal and decrease the inter-subject correlation. We think it is still beneficial to increase the signal-to-noise ratio before the inter-subject correlation analysis. To meet this end, we followed the same two-step procedure in the literature (Pruim et al., 2015).

First, we removed the motion-relevant noise using the ICA-AROMA method. The independent components (ICs) of the BOLD signal were classified as motion-relevant ICs and signal ICs based on their features (e.g., frequency). As these suspected motion-relevant ICs might still include some neuronal components. A safer and more conservative way was to remove their effect "non-aggressively" by putting both motion-relevant components and the signal components into the same GLM and excluding the motion-relevant terms from the BOLD signal. Here, the "motion-relevant terms" were the dot product of motion-relevant components and their estimated beta coefficients. To clear up this misunderstanding, we have clarified this in the manuscript (Line 535-537). Since the signal components were also involved in the same GLM, the shared variance between motion-relevant components and the signal components was preserved.

Second, we further removed the other nuisance variables like WM time series, CSF time series, and signal drift. We acknowledged the risk of reintroducing artifacts into fMRI data during the multi-stage preprocessing (Lindquist et al., 2019). That is why we fitted the CSF timecourse, the WM timecourse, and the quadratic polynomial time trend into the same GLM and regressed out them simultaneously. In the manuscript, we have emphasized this point (Line 543-545). Following Pruum et al. (2015), We did not mix these three components with the GLM in the first step. The timecourses in a conservative mask of CSF or WM and quadratic polynomial time trend are mainly noise. They should be regressed out in an "aggressive" way.

Simony, E., Honey, C. J., Chen, J., Lositsky, O., Yeshurun, Y., Wiesel, A., & Hasson, U. (2016). Dynamic reconfiguration of the default mode network during narrative comprehension. *Nature communications*, 7(1), 1-13.

Pruim, R. H., Mennes, M., van Rooij, D., Llera, A., Buitelaar, J. K., & Beckmann, C. F. (2015). ICA-AROMA: A robust ICA-based strategy for removing motion artifacts from fMRI data. *Neuroimage*, 112, 267-277.

*For contrasting paired ISC values, a permutation test randomly shuffling the group assignments may be a more sensitive (and appropriate) test than using a bootstrap hypothesis test. (Or, equivalently, randomly flipping the signs of the paired contrast across subjects at each permutation; $2^{**}16$ possible permutations). My reasoning here is that this sort of permutation test would accommodate the paired structure of the samples whereas the bootstrap does not. I'm also a bit unsure about allowing for a left-out subject to be randomly sampled into the average of $N - 1$ subjects in a bootstrap sample (line 525). This might make your bootstrap distribution positively skewed (i.e. higher than the original mean leave-one-out ISC), which would result in an overly stringent null distribution (although I'm not sure it would affect contrasts the same way). Same for the ISFC analysis. Note that in our BrainLAK implementation of the bootstrap hypothesis test for leave-one-out ISC, we resample the ISC values themselves (i.e. after computing leave-one-out ISC) rather than resampling the subjects, then computing ISC (<https://github.com/brainiak/brainiak/blob/master/brainiak/isc.py#L650>).*

R: We thank the reviewer again for raising these methodologic concerns. We have clarified these issues in the revised manuscript (Line 603-606; Line 609; Line 654). We hope the revision can clear up the misunderstanding. Below is a more detailed response.

In each bootstrapping iteration, we resampled the participants with replacement, calculated the ISC/ISFC of each condition for this sample, and contrasted the ISC/ISFC among conditions for this sample. Since we resampled the participants and calculated each sample's contrast value rather than resample the participants independently for each condition and then calculated the contrast value, the paired structure should be accommodated. We have emphasized this point in the manuscript (Line 603-606).

The reviewer thought the sampling distribution might shift positively. We suspect the reviewer might think we included the repeated subjects due to resampling with replacement when implementing ISC and ISFC. However, in Line 608 and Line 653, we have pointed out that "Here, 'the other participants' were those excluding him/herself and the repeated ones of him/herself due to resampling with replacement." Thus, the ISC or ISFC was always between participants without any overlap. We have emphasized this point in the manuscript (Line 609 and Line 654).

Minor comments:

When reading the Results, it wasn't obvious to me that subjects were listening to an auditory recording of the stimulus rather than reading a text version of the stimulus (didn't realize this until line 409).

R: We thank the reviewer for pointing out this issue. We have added this information in the last paragraph of the Introduction: "In this study, participants listened to two narrative texts, two argumentative texts, and their corresponding sentence-scrambled versions during the fMRI scanning" (Line 93-95).

Line 173: I would expand this "SES" acronym when you initially introduce it (and point to reference 60)

R: We thank the reviewer for pointing out this error. We have expanded the acronym and added the reference where "SES" first appeared (Line 214-216).

Lines 242, 255, 582: "insignificant" > "nonsignificant"

Line 269: “stimuli-evoked” > “stimulus-evoked”
Line 275: “The argumentative” > “Argumentative”
Line 276: “the functional couplings” > “functional coupling”
Line 285: “after” > “for”
Line 306: “simultaneously featured by two factors” sounds odd to me
Line 326: “yeo” > “Yeo”
Line 327: “unite” > “unit”
Line 359: “that failed to be segmented” > “that segmentation failed”
Line 371: “initialing” > “initial”
Line 548: “The averaged the” > “The averaged”
Line 550: “pair-wised” > “pairwise”
Figure S7: “fsaverag5” > “fsaverage5”
Figure S7: “metric-to-volune” > “metric-to-volume”

We thank the reviewer for pointing out these typos. We have revised them accordingly.

I encourage the authors to share their code (e.g. via GitHub). This seems like a very rich naturalistic dataset, and I hope the authors plan to share it publicly (e.g. via OpenNeuro).

R: We have shared the code for both ISC and ISFC analysis via GitHub (<https://github.com/BottiniLab/NarrativeArgumentative>). We also plan to share the data, pending the approval of the ethical committee of our university.

References:

Lindquist, M. A., Geuter, S., Wager, T. D., & Caffo, B. S. (2019). Modular preprocessing pipelines can reintroduce artifacts into fMRI data. *Human Brain Mapping*, 40(8), 2358–2376. <https://doi.org/10.1002/hbm.24528>

REVIEWERS' COMMENTS:

Reviewer #1 (Remarks to the Author):

The authors' revisions improved the manuscript and I feel that my comments were effectively addressed. The introduction in particular is much stronger now. My only comment is that "pair t" throughout the results section should be "paired t".

Reviewer #2 (Remarks to the Author):

The authors have adequately responded to my initial comments. I have no further issues.

Reviewer #3 (Remarks to the Author):

The authors have improved the framing/motivation of the manuscript with more thorough references to the surrounding literature. They have adequately addressed my comments about directly comparing narrative and argumentative ISC maps (Figure 5a) and demonstrating that within-discourse-type maps are more similar (Figure 5b). The revised manuscript has improved considerably and feels much more complete.